# Structural insights into dsRNA processing by *Drosophila* Dicer-2–Loqs-PD

Shichen Su[1,4], Jia Wang[2,4], Ting Deng[1], Xun Yuan[3], Jinqiu He[1], Nan Liu[2], Xiaomin Li[2], Ying Huang[3], Hong-Wei Wang[2✉] & Jinbiao Ma[1✉]

Small interfering RNAs (siRNAs) are the key components for RNA interference (RNAi), a conserved RNA-silencing mechanism in many eukaryotes[1,2]. In *Drosophila*, an RNase III enzyme Dicer-2 (Dcr-2), aided by its cofactor Loquacious-PD (Loqs-PD), has an important role in generating 21 bp siRNA duplexes from long double-stranded RNAs (dsRNAs)[3,4]. ATP hydrolysis by the helicase domain of Dcr-2 is critical to the successful processing of a long dsRNA into consecutive siRNA duplexes[5,6]. Here we report the cryo-electron microscopy structures of Dcr-2–Loqs-PD in the apo state and in multiple states in which it is processing a 50 bp dsRNA substrate. The structures elucidated interactions between Dcr-2 and Loqs-PD, and substantial conformational changes of Dcr-2 during a dsRNA-processing cycle. The N-terminal helicase and domain of unknown function 283 (DUF283) domains undergo conformational changes after initial dsRNA binding, forming an ATP-binding pocket and a 5′-phosphate-binding pocket. The overall conformation of Dcr-2–Loqs-PD is relatively rigid during translocating along the dsRNA in the presence of ATP, whereas the interactions between the DUF283 and RIIIDb domains prevent non-specific cleavage during translocation by blocking the access of dsRNA to the RNase active centre. Additional ATP-dependent conformational changes are required to form an active dicing state and precisely cleave the dsRNA into a 21 bp siRNA duplex as confirmed by the structure in the post-dicing state. Collectively, this study revealed the molecular mechanism for the full cycle of ATP-dependent dsRNA processing by Dcr-2–Loqs-PD.

In various eukaryotic species, including *Schizosaccharomyces pombe*, worm, fruit fly, plants and human, dicer proteins of the RNase III family enzyme are responsible for producing siRNAs from long dsRNAs. Dicer proteins are also responsible for the production of intrinsic microRNAs (miRNAs) that have essential roles in gene regulation of multiple cellular processes[7,8]. Dicer proteins normally function under the aid or regulation of certain cofactor proteins, more of which are distinct dsRBD proteins, to efficiently produce siRNAs or miRNAs. In *Drosophila*, there are two dicer protein isoforms. Dcr-1 requires Loquacious-PA/PB (Loqs-PA/PB) to produce miRNAs from precursor-miRNAs (pre-miRNAs). Dcr-2 requires Loqs-PD to produce siRNAs from dsRNAs and another double-stranded RNA-binding domain (dsRBD) protein R2D2 to load siRNAs into Argonaute2 (Ago2)[1].

Previous structural studies suggest that canonical dicer proteins usually contain three modules: the N-terminal bottom module, the C-terminal core module and the cap module in the middle of the protein sequence[9,10]. The core module comprises two RNase III domains—RNase IIIa and RNase IIIb, which are responsible for cleaving the duplex stem of dsRNAs—and has an additional one or two dsRBDs in the C-terminal region. The cap module can specifically recognize 3′-end two-nucleotide overhang and 5′-phosphate through the PAZ domain and the Platform domain, respectively. The bottom module comprises a RIG-I-like helicase domain and a dsRBD-like DUF283 domain. Although the RIG-I-like helicase domains are highly conserved in dicer proteins, only those that process long dsRNAs to produce consecutive siRNAs are ATP-dependent, such as *Drosophila* Dcr-2. By contrast, the human Dicer and *Drosophila* Dcr-1 proteins do not have ATP-hydrolysis activity in their helicase domains. Previous studies demonstrated that the helicase domain of *Drosophila* Dcr-2 is also required to discriminate between the dsRNA termini for specific cleavage[11,12].

Several structures of dicer proteins from different species have been reported in recent years, including human Dicer–TRBP in complex with a pre-miRNA substrate[9], *Arabidopsis* DCL1 and DCL3 in complex with their substrates[13,14], and *Drosophila* Dcr-2 in complex with dsRNA[12]. However, the molecular mechanism of a full cycle of RNA substrate processing by dicer remains unclear, especially that by ATP-dependent dicer proteins. Here six cryo-electron microscopy (cryo-EM) structures of *Drosophila* Dcr-2–Loqs-PD in apo and in complex with dsRNA substrate in the absence or in the presence of ATP were captured, illuminating the continuous conformational changes of Dcr-2–Loqs-PD after binding to dsRNA and translocating until forming the active dicing state and cleaving the dsRNA substrate into an siRNA duplex in the post-dicing state.

[1]State Key Laboratory of Genetic Engineering, Collaborative Innovation Center of Genetics and Development, Department of Biochemistry and Biophysics, School of Life Sciences, Fudan University, Shanghai, China. [2]Ministry of Education Key Laboratory of Protein Sciences, Tsinghua-Peking Joint Center for Life Sciences, Beijing Advanced Innovation Center for Structural Biology, Beijing Frontier Research Center of Biological Structures, School of Life Sciences, Tsinghua University, Beijing, China. [3]Shanghai Key Laboratory of Biliary Tract Disease Research, Shanghai Research Center of Biliary Tract Disease, Department of General Surgery, Xinhua Hospital, Shanghai Jiao Tong University School of Medicine, Shanghai, China. [4]These authors contributed equally: Shichen Su, Jia Wang. ✉e-mail: hongweiwang@tsinghua.edu.cn; majb@fudan.edu.cn

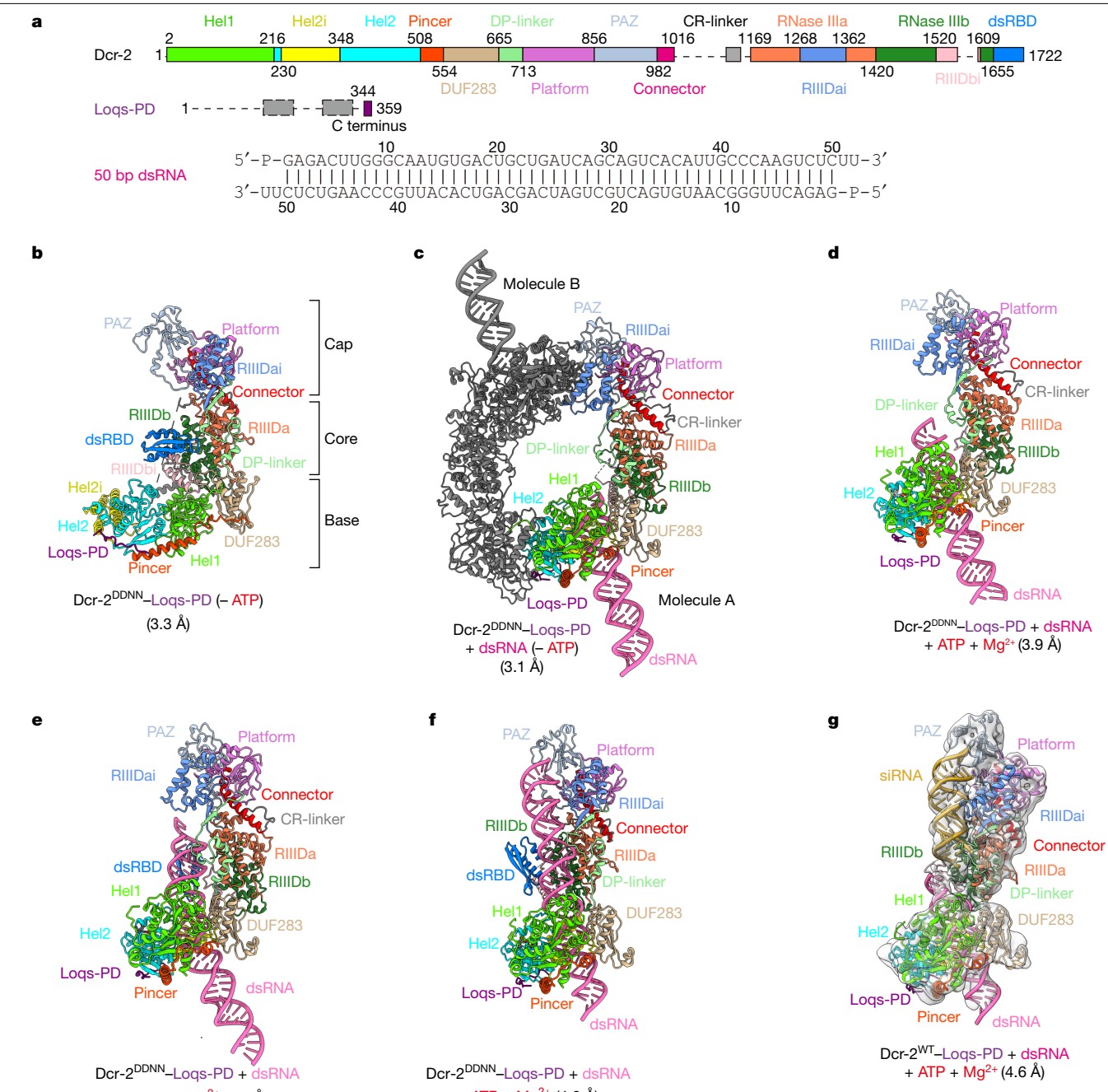

**Fig. 1 | Cryo-EM structures of Dcr-2–Loqs-PD in different dsRNA-binding states. a**, Schematic of Dcr-2, Loqs-PD and 50 bp dsRNA. Unless specified otherwise, the colour scheme of Dcr-2 and Loqs-PD is used throughout all of the figures. **b**, Cartoon model of Dcr-2–Loqs-PD in the apo state. The cap, core and base modules are labelled on the right. **c**, Cartoon model of Dcr-2–Loqs-PD–50 bp dsRNA dimer in the initial dsRNA-binding state (initial state). Molecule A is coloured, molecule B is shown in grey. **d**,**e**, Cartoon model of Dcr-2–Loqs-PD–50 bp dsRNA complex in the early-translocation (**d**) and mid-translocation (**e**) state. **f**,**g**, Cartoon model of Dcr-2–Loqs-PD–50 bp dsRNA complex in the active dicing (**f**) and post-dicing (**g**) state. The siRNA in **g** is coloured in goldenrod. The post-dicing state is shown as a transparent cryo-EM map. The model in **c**–**g** is aligned by base region. The components and resolution of each state are labelled below.

## Overall structures

We used single-particle cryo-EM to study the structure and mechanism of *Drosophila* Dcr-2 in the dsRNA-processing cycle. We purified recombinant wild-type (WT) Dcr-2 and its catalytically inactive mutant Dcr-2(D1217N/D1476N) (hereafter Dcr-2(DDNN)), corresponding to *Aquifex aeolicus* RNaseIII (*Aa*RNaseIII) D44N[15], from insect cells, and mixed the Dcr-2 constructs with purified full-length Loqs-PD to form stable WT Dcr-2–Loqs-PD and Dcr-2(DDNN)–Loqs-PD complexes, respectively

(Extended Data Fig. 1). We also designed and purified a 50 bp palindromic sequence dsRNA with a 3′ two-nucleotide overhang and a 5′ monophosphate terminus as the substrate on the basis of previous studies[11,16] (Fig. 1a). Using a different combination of assembly components and buffers with or without ATP, we solved a series of cryo-EM structures of Dcr-2–Loqs-PD during the dsRNA-processing cycle at resolutions ranging from 3.1 Å to 4.6 Å. These structures include the Dcr-2(DDNN)–Loqs-PD complex in the apo state and the dsRNA initial binding state without ATP; the dsRNA early-translocation and mid-translocation states and the active

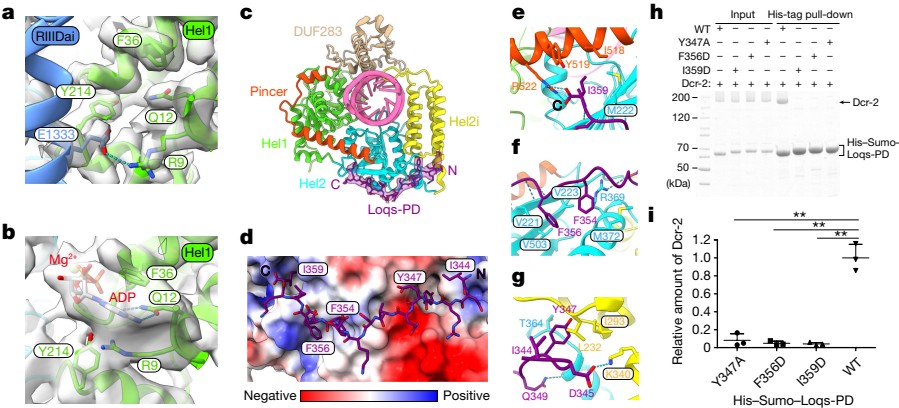

**Fig. 2 | The ATP-binding pocket and interactions between helicase domain and C terminus of Loqs-PD. a**, Interactions between the ATP-binding site and RIIIDai from molecule B in the initial binding state. **b**, Details of the ATP-binding site in the early-translocation state. **c**, Overview of the Loqs-PD binding sites in the helicase domain of Dcr-2. Loqs-PD is shown as a transparent cryo-EM map. **d**, Surface presentation of the Loqs-PD binding region in the helicase domain. Loqs-PD is shown as a stick model. The surface is coloured by electrostatic potential. **e–g**, Details of the interactions of Dcr-2 with Ile359 (**e**), Phe354 and

Phe356 (**f**), and Tyr347 (**g**) regions of C terminus of Loqs-PD. **h**, SDS–PAGE gels of the pull-down result for Dcr-2 with the WT and three variants of His-tagged Loqs-PD. Gel source data are provided in Supplementary Fig. 1. **i**, Quantification of the pull-down data in **h**. Data are mean ± s.d. $n = 3$ biologically independent experiments. Statistical analysis was performed using two-sided paired $t$-tests; **$P < 0.01$. The exact $P$ values for WT versus Y347A, F356D and I359D are 0.0037, 0.0063 and 0.0091, respectively.

dicing state in the presence of ATP; and Dcr-2$^{WT}$–Loqs-PD complex in its post-dicing state (Fig. 1b–g and Extended Data Figs. 2 and 3).

The cryo-EM structure of Dcr-2–Loqs-PD in the apo state at a resolution of 3.3 Å shows that Dcr-2 has the same L-shaped domain organization as human Dicer[9], but with more structural elements solved (Fig. 1b). Similar to human Dicer, the N-terminal helicase region forms the base of the L-shaped structure and the DUF283 domain is located at the corner of the L and just below the catalytic active core. The previously unmodelled linker between the DUF283 and Platform domains (DP-linker) in human Dicer is now well resolved in Dcr-2 as a structural element that interacts with the RNase IIIa domain through two α-helices and one β-strand. The Platform and PAZ domains surrounding the Connector helix sit at the top of the core processing region. The intrinsic RNase III dimer comprising the RIIIDa and RIIIDb domains is located in the centre of the molecule with the insertion domains (RIIIDai and RIIIDbi) sitting in symmetric positions (Extended Data Figs. 4 and 5a–e). The C-terminal dsRBD interacts with the DP-linker and shields the RNase-activity centre (Extended Data Fig. 5e).

Compared with the apo state, the overall structures of Dcr-2–Loqs-PD complex undergo major conformational changes when interacting with the dsRNA substrate, especially on the helicase and DUF283 domains (Extended Data Fig. 6a–e). By contrast, the structures in the active dicing and post-dicing states demonstrate further conformational changes in the PAZ–Platform module. These structures and their mechanistic insights are discussed in more detail in the sections below.

## Interaction between Loqs-PD and Dcr-2

In the absence of ATP, we found that Dcr-2–Loqs-PD tends to dimerize and forms an initial binding complex with dsRNA (Fig. 1c). The dimerization interface involves the RIIIDai domain of one Dcr-2 and the Hel2 domain of the other Dcr-2, and mutation of residues in the dimerization interface results in dissociation of dimer (Fig. 1c and Extended Data Fig. 5f–k). The dimerization interface also disrupts the ATP-binding site of the helicase domain (Fig. 2a) that otherwise is well maintained to bind to ADP/ATP in the translocation and dicing states (Fig. 2b). The high-resolution (3.1 Å) cryo-EM structure of the initial binding complex is good enough for us to unambiguously identify the density of the C-terminal tail (amino acids 344–359) of Loqs-PD interacting with the helicase domain of Dcr-2 (Fig. 2c and Extended Data Fig. 6f,g). The other portions of the Loqs-PD protein cannot be distinguished in the cryo-EM maps probably due to its flexible nature. The short

C-terminal tail extends and wraps over the surface of the Hel2i, Hel2 and Pincer domains (Fig. 2c,d), mainly through hydrophobic interactions and hydrogen bonds (Fig. 2e–g). The C-terminal end residue of Loqs-PD, Ile359, is accommodated by the hydrophobic pocket comprised by Met222 from Hel2, Ile518 and Tyr519 from Pincer. Furthermore, the C-terminal carboxyl group of Ile359 of Loqs-PD forms a salt bridge interaction with the side chain of Arg522 from Pincer (Fig. 2e). Two hydrophobic residues identified previously[17], Phe354 and Phe356 of Loqs-PD, bind to the hydrophobic patch on the Hel2 domain, whereas the aromatic ring of Phe356 stacks with the side chain of Arg369 (Fig. 2f). Tyr347 of Loqs-PD inserts into a hydrophobic cage comprising Leu232, Ile293 and Thr364 right at the interface of the Hel2 and Hel2i domains (Fig. 2g). Mutations to any of the three hydrophobic residues (Y347A, F356D, I359D) of Loqs-PD notably reduced its interaction with Dcr-2 as shown in the pull-down assay (Fig. 2h,i), indicating their essential roles in Dcr-2–Loqs-PD interactions.

## Conformational changes after dsRNA binding

dsRNA binding to Dcr-2 induces a large conformational change of the helicase and DUF283 domains (Supplementary Videos 1 and 2). The N-terminal helicase domain of Dcr-2 belongs to the superfamily 2 group of helicase proteins, like RIG-I-like receptors (RLRs), and contains four subdomains: Hel1, Hel2i, Hel2 and Pincer[18]. However, in contrast to RLRs, of which the Pincer subdomain links the helicase domain to the protein's C-terminal domain, the Pincer subdomain of Dcr-2 links the helicase domain to the DUF283 domain. After binding to a dsRNA substrate, the helicase domain together with the DUF283 domain transform from a stretched conformation (Fig. 3a) to a closed one (Fig. 3b), wrapping around the RNA helix (Supplementary Video 2). Notably, the DUF283 domain rotates about 150° around Hel1 and directly contacts the dsRNA (Fig. 3c). Meanwhile, the second α-helix of Pincer rotates together with DUF283, forming the traditional V-shaped structure and clamping Hel1 (Fig. 3b,c). These conformational changes reorient the DUF283 domain between Hel1 and Hel2i and tightly interact with the two subdomains (Extended Data Fig. 6h,i).

## Recognition of dsRNA by helicase–DUF283

The near-atomic high resolution (3.1 Å) structure of the initial binding state enabled us to examine the interactions between the

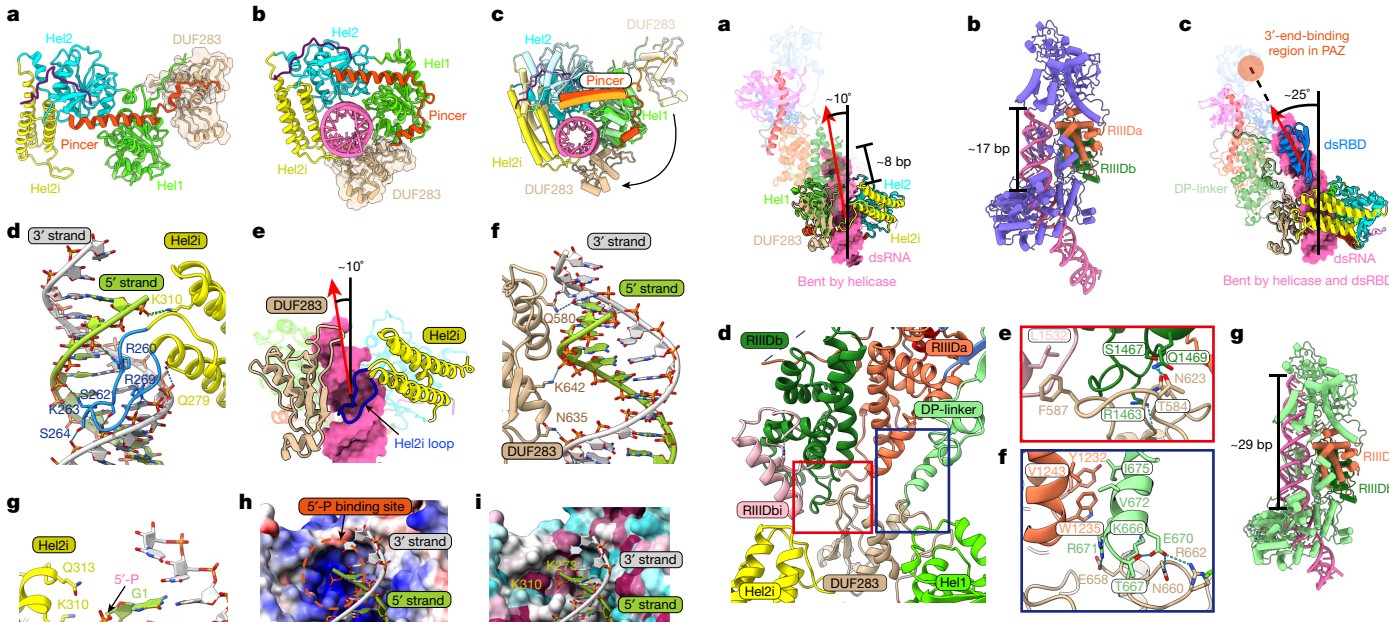

**Fig. 3 | Substantial conformational changes of the helicase and DUF283 domains induced by dsRNA binding and specific dsRNA recognitions.** **a,b**, Overview of the helicase domain in the apo state (**a**) and the initial binding state (**b**). DUF283 is presented as transparent surface. The structures in **a** and **b** are aligned by the Hel1 domain. **c**, Superimposition of the helicase domain aligned by the Hel1 domain in the apo and initial binding states. The apo state is coloured with a similar but lighter colour compared with the initial binding state. The movement of DUF283 is labelled by the black arrow. **d**, The close-up view of the Hel2i–dsRNA interface. The special loop is coloured in dodger blue. The 5′ and 3′ strands of dsRNA are shown in lime green and light grey, respectively. **e**, Overview of DUF283 and Hel2i interacting with dsRNA in the initial binding state. The special loop of Hel2i (257–271) is coloured in blue. dsRNA is shown as surface. The axis direction of dsRNA termini bound by the helicase domain is shown by the red arrow, and the black line represents the axis of the free dsRNA. The bending angle is labelled at the top. **f**, Magnified view of the DUF283–dsRNA interface. dsRNA is coloured as in **e**. **g**, Details of dsRNA terminal recognition of Hel2i domain. **h,i**, The surface presentation of the helicase domain around the terminal binding region. The surface of the helicase domain is coloured by electrostatic potential (**h**) and sequence conservation (**i**). The 5′-phophate-binding site is marked by an orange dotted circle. The two residues involved in 5′-phosphate recognition are labelled in **i**.

**Fig. 4 | The relative rigid structures of the Dcr-2–Loqs-PD complex and further dsRNA bending in the translocation states. a**, Distortion of dsRNA in the early-translocation state. The dsRNA is presented in surface mode. Dcr-2–Loqs-PD is presented in cartoon mode. The translocation length of dsRNA (~8 bp) compared with the initial binding state is labelled. Domains interacting with dsRNA are opaque with a black outline. The axis of ideal A-form dsRNA extended from outside the helicase domain is shown by the black line. The helical axis of the helicase domain-bound dsRNA is shown by the red arrow. **b**, Overview of the mid-translocation state. The translocation length of dsRNA (~17 bp) is labelled in **b** as in **a**. The transparent cryo-EM map is shown in **c**. **c**, Distortion of dsRNA in the mid-translocation state is shown in the same mode as in **a**. The 3′-end binding region of PAZ in the mid-translocation state is labelled by the transparent orange circle. The axis of ideal A-form dsRNA extended from outside the helicase domain is shown by the black line. The helical axis of C-terminal dsRBD domain-bound dsRNA is shown by the red arrow. **d**, Overview of the interdomain contacts between DUF283 and the core region in the initial binding state. **e**, Magnified view of the interdomain contacts between the DUF283 and RIIIDb–RIIIDbi domains, corresponding to the orange-red box in **d**. **f**, Magnified view of the interdomain contacts between DUF283 with RIIIDa through the DP-linker, corresponding to the dark-blue box in **d**. **g**, Overview of the pre-dicing model. Model of the pre-dicing state based on the mid-translocation state with further RNA translocation. The translocation length of dsRNA (~29 bp) is labelled as in **a**.

helicase–DUF283 module and the dsRNA with precision and unambiguity at the residue level (Extended Data Fig. 7a). The dsRNA-recognition mode of the Hel1 and Hel2 subdomains is conserved with the RLR (Extended Data Fig. 7). Distinctly, a loop specific for the Dicer Hel2i subdomain forms intimate contacts with the dsRNA substrate, which corresponds to the second helix in Hel2i of RLRs, extending into the major groove of dsRNA duplex (Fig. 3d and Extended Data Fig. 7d,f). Residues in this loop extensively interact with the backbone phosphates of the major groove that is widened subsequently (Fig. 3d), resulting in the dsRNA duplex bending around 10° towards the DUF283 domain (Fig. 3e). Moreover, the phosphate group of the 5′ strand terminus of the dsRNA is specifically recognized by Hel2i through a salt-bridge interaction between 5′-phosphate and Lys310 of Hel2i (Fig. 3d,g). The electrostatic potential surface suggests that there is a basic and conserved pocket around the 5′-phosphate of dsRNA in the Hel2i domain (Fig. 3h,i). Furthermore, the DUF283 domain recognizes the minor and major groove of the dsRNA terminal duplex through Gln580 from region 1 and Asn635/Lys642 from region 3 of the dsRBD fold, respectively (Fig. 3f). In summary, the conformational changes of the helicase

and DUF283 domains induced by dsRNA binding are coupled to the simultaneous recognitions of the major and minor groove of the dsRNA terminal duplex and the 5′-phosphate by Hel2i and DUF283, determining the initial binding of dsRNAs to the specific terminal duplex feature.

## Structural states during translocation

Recent single-molecule analysis demonstrated that Dcr-2 translocates along the dsRNA after its helicase initially binds to the dsRNA terminus in the presence of ATP[19]. To capture the different steps during the translocation of Dcr-2–Loqs-PD along the dsRNA, we used cryo-EM to analyse the sample of Dcr-2[DDNN]–Loqs-PD in complex with the dsRNA substrate under conditions with ATP and Mg[2+]. Through single-particle classification, we captured two representative cryo-EM structures of Dcr-2–Loqs-PD in complex with the dsRNA substrate in translocation states (Fig. 1d,e and Extended Data Fig. 3). In the first captured structure, termed the early-translocation state, about 8 bp of duplex threads through the helicase domain towards the catalytic centre of Dcr-2 (Fig. 1d). In comparison to the initial binding state (Fig. 1c), the

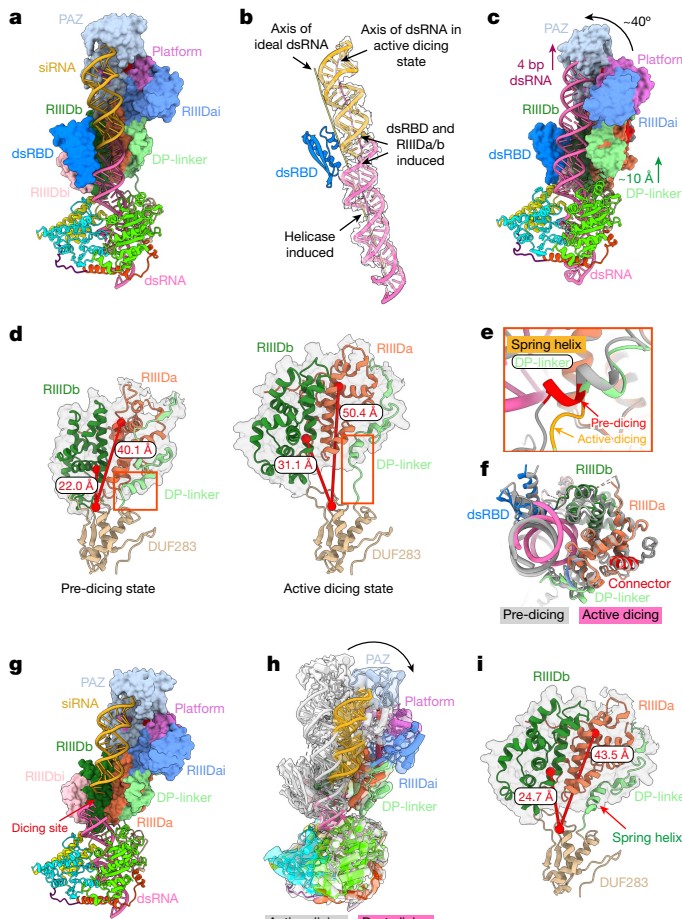

**Fig. 5 | The conformational changes associated with transitions from the pre-dicing to active dicing and to post-dicing states. a**, Overview of the Dcr-2–Loqs-PD–50 bp dsRNA complex in the active dicing state. Dcr-2 (672–1722) is shown in surface mode. The siRNA region is coloured in gold. **b**, The distortion of dsRNA in the active dicing state. The axes of free dsRNA and active dicing dsRNA are coloured in yellow and hot pink, respectively. The torsion position induced by Dcr-2 is labelled. **c**, Overview of the pre-dicing model. The display mode is the same as in **a**. The conformational change from the pre-dicing state (**c**) to the active dicing state (**a**) is marked by purple, green and grey arrows in **c**. **d**, The distance between Cα atoms of Q580 from DUF283 and both RIIIDs (E1213 and E1472) in the pre-dicing (mid-translocation) and active dicing states. Cα atoms are labelled as spheres. RIIIDs are shown as transparent surface. The conformational change of DP-linker is marked by the orange-red box.

**e**, Magnified view of the spring helix α1 of the superimposed structures in the mid-translocation state (grey) and the active dicing state (coloured) aligned by RIIIDs, corresponding to the orange-red box in **d**. The first half of α1 is coloured in red for the mid-translocation state and orange for the active dicing state, respectively. **f**, Cross-section of the processing centre of the superimposed mid-translocation state (grey) and active dicing state (coloured) aligned by RIIIDs. **g**, Overview of the post-dicing state. The display mode is the same as in **a** and **c**. The break in the dsRNA is labelled by the red arrow. **h**, Superposition of the structures in the active dicing and post-dicing state shown as a transparent cryo-EM map. The conformational change from the active dicing state to the post-dicing state is labelled by the black arrow. **i**, The distance between Gln580 of DUF283 and both RIIIDs (Glu1213 and Glu1472) in the post-dicing state as in **d**. The spring helix α1 is marked by orange-red arrow.

Dcr-2–Loqs-PD complex in the early-translocation state has almost no conformational changes and no additional interactions with dsRNA, except that the cap and core modules shift about 3 Å away from the dsRNA (Extended Data Fig. 8a,b and Supplementary Video 1). The trajectory of dsRNA in the early-translocation state is same as in the initial binding state (Fig. 4a and Extended Data Fig. 8c). We captured another structure in the mid-translocation state, in which about 17 bp of a dsRNA duplex threads through the helicase domain further towards the catalytic centre of Dcr-2 (Fig. 4b). When superimposed onto structures of the initial binding and early translocation states, the cap–core modules sway back about 2 Å towards the dsRNA (Extended Data Fig. 8a,b and Supplementary Video 1). Importantly, in the mid-translocation state, the C-terminal dsRBD of Dcr-2 appears in the map and interacts with the dsRNA substrate (Extended Data Fig. 8d), resulting in further bending of the axis of the dsRNA about 25°, and the trajectory of the dsRNA is directly towards the PAZ domain (Fig. 4c and Supplementary Video 1). Removal of the C-terminal dsRBD of Dcr-2 completely abolished the cleavage ability of the enzyme on the dsRNA substrate and cryo-EM

analysis revealed no further bending of the dsRNA during the translocation by the helicase domain (Extended Data Fig. 9), underlining the critical role of C-terminal dsRBD in ATP-dependent dsRNA processing by Dcr-2. During the translocation, the interactions between the DUF283 domain and RIIIDa/b domains (Fig. 4d), including relatively flexible interactions between the DUF283 and RIIIDb/RIIIDbi domains (Fig. 4e), and relatively rigid interactions between the DUF283 domain and RIIIDa through the DP-linker domain (Fig. 4f and Extended Data Fig. 8f), remain almost the same as in the initial binding state (Extended Data Fig. 8a). This suggests that the overall domain configuration of Dcr-2 is relatively rigid during the translocation process until the dsRNA terminus reaches the Platform–PAZ domains.

A comparison of the cryo-EM structures of the early- and mid-translocation states also demonstrates a very similar structure in their helicase domains, of which the ATP-hydrolytic pockets have an EM density for either ATP or ADP-Mg$^{2+}$ (Fig. 2b and Extended Data Fig. 8e). The current resolution is yet not high enough to distinguish the nucleotide state or separate the ATP-hydrolytic steps in the complex.

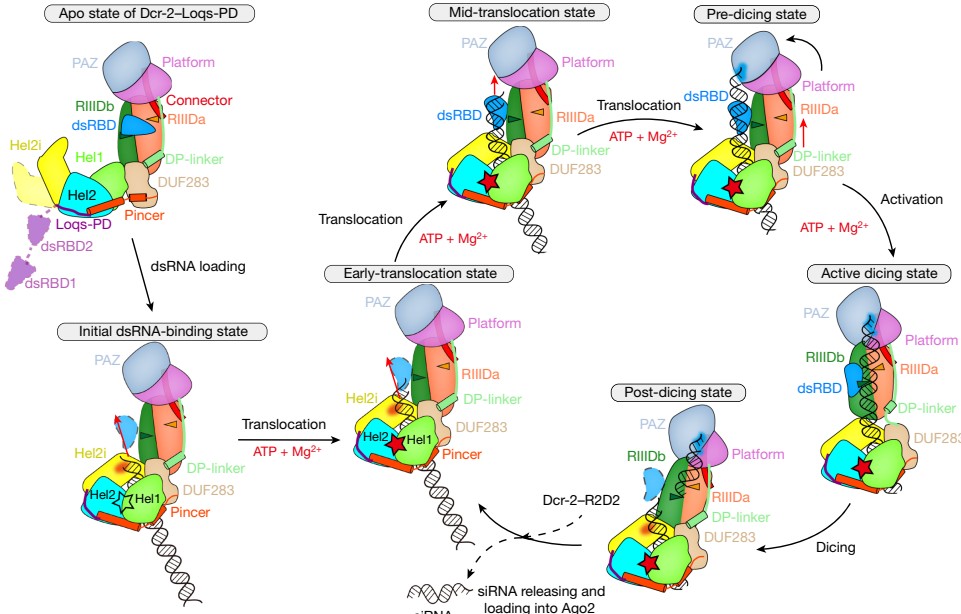

**Fig. 6 | The model of the Dcr-2–Loqs-PD complex in the dsRNA-processing cycle.** The stars represent the mature ATP-binding site, with red and hollow representing whether or not ATP or ADP is bound. The sky-blue region in the Platform–PAZ domains is the dsRNA terminal recognition site. Domains shown in transparent mode with a dotted outline are not visible in that state. The triangles in RIIIDs represent the RNase active centre. The red arrows show the translocation direction.

However, the translocation of the dsRNA's terminus threading through the helicase domain towards the RNase active centre of Dcr-2 is clearly ATP driven, as none of the translocation intermediate states were observed in the sample without ATP (only the apo state or initial binding state were observed). It is expected that a deeper translocated state has no further conformational changes until the dsRNA terminus is bound by the Platform–PAZ domains. Thus, on the basis of the structure of mid-translocation state, we modelled the structure of a fully translocated state by threading a further 12 bp dsRNA through the helicase domain compared with the mid-translocation state (Fig. 4g and Supplementary Video 1), which is similar to the recently reported structures of *Arabidopsis thaliana* DCL-1 (*At*DCL-1) in complex with a primary miRNA substrate[13] (Extended Data Fig. 8g). However, when superimposing the modelled structure of the fully translocated state onto the active dicing structure of *At*DCL3 in complex with a dsRNA substrate (Protein Data Bank (PDB): 7VG2) by the RIIIDa/b domains[14], the interaction between the DUF283 and RIIIDb domains (Fig. 4d,e) has steric clashes with the dsRNA bound to the RNase active centre of RIIIDa/b of *At*DCL3 in the active dicing state (Extended Data Fig. 8h), suggesting that this interaction would prevent the dsRNA substrate from accessing the RNase active centre of the RIIIDa/b domains during the translocation of Dcr-2 along dsRNA. This modelled structure is designated the pre-dicing state (Fig. 4g).

### The structure of the active dicing state

We wondered how Dcr-2 is activated from the pre-dicing state to allow the dsRNA to reach the active centre of RIIIDa/b. We captured a structure of Dcr-2–Loqs-PD in complex with 50 bp dsRNA in an active dicing state under the same conditions as for the translocation states (Figs. 1f and 5a and Extended Data Fig. 3), in which the terminus of dsRNA was bound by the Platform–PAZ domains through recognition of the 3′ two-nucleotide overhang and 5′-phosphate, respectively (Extended Data Fig. 8i). The dsRNA is held in the RNase active centre formed by RIIIDa and RIIIDb, and two metal ions are bound in the two catalytic sites (Extended Data Fig. 8j), suggesting that the structure is in the active dicing state. As the Dcr-2 used here is a double mutant that has no dicing activity, the complex halts in a stable state, of which the structure was captured at a resolution (4 Å)—higher than the resolution of the structure in the mid-translocation state (4.2 Å). The C-terminal dsRBD in the active dicing state is also better defined for its interactions with the minor and major grooves of dsRNA mainly by residues of Arg1658, Lys1706 and Lys1702 (Extended Data Fig. 8k,l). The axis of dsRNA bound by the Dcr-2–Loqs-PD complex in the active dicing state is obviously distorted due to the interactions of dsRBD and RIIIDa/b (Fig. 5b). Moreover, compared with the structure in the pre-dicing state, the dsRNA in the active dicing state further threads about 4 bp through the helicase domain (Fig. 5a,c), resulting in the cap and core modules transforming about 10 Å away from the bottom module and rotating 40° (Fig. 5a,c and Supplementary Video 1). The keys for the conformational changes are the disruption of interactions between the DUF283 and RIIIDb domains (Fig. 5d), and the stretch of DP-linker in which a helix unwound about one turn (Fig. 5e and Supplementary Video 1), eliminating the misalignment between the bottom and cap–core modules of Dcr-2 and allowing the dsRNA duplex to shift about 10 Å (Fig. 5f) and position exactly in the catalytic active centre of the RIIID domains (Extended Data Fig. 8j). These conformational changes suggest that the ATP hydrolysis by the helicase domain is converted into a tension accumulated during the translocation of Dcr-2 on dsRNA, driving the transformation of the pre-dicing state to the active dicing state.

### siRNA releasing in post-dicing state

To investigate the dynamic dicing process of Dcr-2, we mixed WT Dcr-2–Loqs-PD with a 50 bp dsRNA substrate, then added ATP and Mg²⁺ just before generating the cryo-EM samples. From such an enzymatically active sample, we were able to capture a complex structure in the post-dicing state at a relatively lower resolution (4.6 Å) (Figs. 1g and 5g). In this structure, a clear breakage of the dsRNA after the dicing near to the catalytic centre was observed (Fig. 5g) and exactly 21 bp away from the PAZ-domain-binding terminus (Extended Data Fig. 8m). The cleavage of dsRNA disrupts the binding site for the C-terminal dsRBD in the active dicing state (Fig. 5b and Extended Data Fig. 8k–m), and probably results in the dsRBD becoming more flexible and losing its density

in the averaged EM map of the post-dicing state (Fig. 5g,h), which may be also favourable to the release of cleaved siRNA products (Extended Data Fig. 8m and Supplementary Video 1). Compared with the structure in the active dicing state, the cap–core module with siRNA in the post-dicing state rotated about 20° (Fig. 5h), which is derived from the partial restoration of the spring helix in the DP-linker, probably accompanied by the release of the tension (Fig. 5i and Supplementary Video 1). The remaining dsRNA duplex bound by the helicase domain also returns to a conformation similar to the early-translocation state (Extended Data Fig. 8n), which enables the translocation of Dcr-2–Loqs-PD to enter the next processing cycle (Fig. 6 and Supplementary Video 1).

## Discussion

Dcr-2 is a Dicer protein that requires ATP hydrolysis to process long dsRNA and produce siRNA duplex in vivo[20,21], although later studies showed that human Dicer does not require ATP to cleave the dsRNA in vitro[22,23]. Loqs-PD is a cofactor protein that comprises two dsRBDs that are responsible for recruiting siRNA precursor substrates for Dcr-2. The interactions between the C-terminal tail of Loqs-PD and Dcr-2 provide a structural basis for the specific requirement of Loqs-PD for endo-siRNA production[3]. Its strong interaction with the Dcr-2 helicase domain lends a proximate convenience for loading siRNA precursors onto the helicase domain for initial binding. The two dsRBDs increase the dsRNA-binding affinity of the helicase, making it easier for the complex to reach the initial binding state (Extended Data Fig. 10a,b). However, the dsRBDs were not observed in the density of the two-dimensional average of the initial binding state, suggesting that the dsRBDs of Loqs-PD have only an assisting role in the initial dsRNA binding of the helicase domain of Dcr-2, which is consistent with the single-molecule result from a previous study[19].

It is well known that Dcr-2 is an ATP-dependent enzyme. However, we found that Dcr-2 forms a homodimer when binding to dsRNA a substrate in the absence of ATP. Although we cannot conclude whether the dimerization is an in vitro artifact or whether it is physiologically relevant, this finding helped us to solve near-atomic high-resolution structures of Dcr-2–Loqs-PD in the apo and initial binding states under conditions without ATP. Interestingly, the conformational changes of the helicase and DUF283 domains induced by dsRNA binding, which are similar to RLRs, do not require ATP hydrolysis, instead, forming an ATP-binding pocket (Fig. 2a). Although the translocation requires ATP hydrolysis, Dcr-2–Loqs-PD does not undergo major conformational changes. However, in comparison to the modelled pre-dicing state, substantial conformational changes in the dicing state accompanied by an additional 4 bp dsRNA threading through helicase domain (Fig. 5) suggested that the transformation of Dcr-2–Loqs-PD into the active dicing state requires ATP hydrolysis. Thus, both the translocation and activation of Dcr-2–Loqs-PD are ATP-dependent (Fig. 6). Note that the helicase domain is responsible for reducing non-specific cleavage, indicating that the ATP-dependent translocation of dsRNA is the key property of Dcr-2 for controlling the precision of RNA processing (Extended Data Fig. 10c).

The DUF283 domain is a distinct domain that is found only in higher eukaryotic dicer proteins. The bioinformatic[24] and structural[25] studies suggested that DUF283 has a non-canonical dsRBD fold, and functions in protein–protein interactions[25] and dsRNA annealing[26]. Our high-resolution structures of Dcr-2–Loqs-PD in the apo and initial binding states, as well as the translocation and dicing states, revealed that DUF283 has a central role in conformational changes: from the apo to initial binding state and from the pre-dicing to active dicing state. In the transition from the apo to the initial binding state, the DUF283 domain undergoes conformational changes in a manner similar to the C-terminal domain in RLRs (Fig. 3). Moreover, DUF283 interacts with RIIIDb and RIIIDa, blocking the access of dsRNA to the catalytic centre of RIIIDa/b (Extended Data Fig. 8h) and preventing non-specific

cleavage of dsRNA during the translocation. In the active dicing state, the interaction between DUF283 and RIIIDb is disrupted (Fig. 5d) and the DP-linker is stretched (Fig. 5e), allowing the dsRNA substrate to enter the active centres of RIIIDa/b for precise cleavage (Fig. 5f).

Recent research suggested that ATP is required for recognition and discrimination of dsRNA termini[12,27]. We have identified a 5′-phosphate-binding pocket in Hel2i in the initial binding state (Fig. 3g–i) under conditions without ATP. There are extensive studies on the 5′-triphosphate recognition by the C-terminal domain of RIG-I that has similar conformational change of helicase after dsRNA binding[28,29]. Dcr-2 may sense the 5′-triphosphate of RNAs using a similar binding pocket to that demonstrated previously[12,27]. Recent single-molecule analysis[19] supports the conclusion from our structures in this study that terminal loading of dsRNA as the initial step and translocation along the dsRNA are required to activate the processive cleavage activity of Dcr-2.

## Summary

In conclusion, we used cryo-EM to solve the structures of Dicer-2 in complex with its cofactor Loqs-PD in its apo and initial dsRNA-binding states in the absence of ATP, and in multiple translocation and dicing states in the presence of ATP, deciphering the multiple ATP-dependent conformational changes during the full cycle of dsRNA processing by Dcr-2–Loqs-PD from initial dsRNA binding to dicing of the dsRNA substrate into the siRNA duplex (Fig. 6 and Supplementary Video 1).

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

# Methods

## Data reporting

No statistical methods were used to predetermine sample size. The experiments were not randomized and the investigators were not blinded to allocation during experiments and outcome assessment.

## Protein expression and purification

The gene encoding full-length *Drosophila melanogaster* Dcr-2 (UniProt: A1ZAW0) was cloned from the recombinant pFastBac-Dcr-2 plasmid (gifted by the Q. Liu laboratory). Full-length *Dm*Loqs-PD (UniProt: M9MRT5) was PCR amplified from *Drosophila* cDNA and cloned into a modified pET28a (with a 6×His-SUMO tag). The constructs of WT *Dcr-2*, *Loqs-PD* and other mutations were generated using a standard PCR-based cloning strategy and cloned into the corresponding vectors, and their identities were confirmed by sequencing analysis.

*Dcr-2* or its mutants was expressed using the Bac-to-Bac baculovirus expression system (Invitrogen) in sf9 cells at 27 °C. One litre of cells ($2 \times 10^6$ cells per ml, medium from Expression Systems) was infected with 20 ml baculovirus at 27 °C. After growth at 27 °C for 48 h, the cells were collected, resuspended in buffer A (150 mM NaCl, 20 mM Tris-HCl pH 8.0, 10% glycerol, 20 mM imidazole) with 0.5 mM PMSF and protease inhibitors, and lysed by adding 0.5% Triton X-100 and shaken gently for 30 min at 4 °C. Dcr-2 was purified to homogeneity using Ni-NTA affinity, Hitrap Q column (Cytiva), 2nd Ni-NTA affinity and size-exclusion chromatography using the Superdex 200 10/300 Increase column (Cytiva) (in that order).

*Loqs-PD* and its mutants were expressed in *Escherichia coli* BL21 (DE3). Loqs-PD was first purified by Ni-NTA affinity chromatography. Using protease ULP1 to remove the 6×His–SUMO tag, and dialysis was applied to remove imidazole. The sample was then applied to a second Ni-NTA chromatography and the flow-through was collected for size-exclusion chromatography using the Superdex 200 16/600 column (Cytiva). Fractions corresponding to the apo Loqs-PD were collected and concentrated to about 10 mg ml⁻¹.

## Preparation of dsRNAs

The dsRNAs were in vitro transcribed using T7 RNA polymerase. The pUC19 plasmids containing target sequences with 3′-HDV ribozyme sequences were linearized by EcoRI, extracted with phenol–chloroform and precipitated with isopropanol. The in vitro transcription reaction was performed at 37 °C for 5 h in the buffer containing 100 mM HEPES-K (pH 7.9), 10 mM MgCl₂, 10 mM dithiothreitol (DTT), 6 mM NTP each, 2 mM spermidine, 200 µg ml⁻¹ linearized plasmid and 100 µg ml⁻¹ T7 RNA polymerase. For the 5′-monophosphate RNA, 40 mM GMP was added in the transcription reactions. EDTA at a final concentration of 20 mM was added to the samples containing palindromic transcripts. The samples were heated to 95 °C for 5 min and then slowly cooled to room temperature. The annealed transcripts were purified by 8% denaturing urea PAGE, eluted from gel slices and precipitated with isopropanol. After centrifugation, the RNA precipitant was collected, washed twice with 70% ethanol and air-dried, and the RNA was dissolved in ultrapure water. We next used T4 PNK (NEB, M0201) to remove the 2′,3′ cyclic phosphate at the 3′ end of the RNA. The FAM-labelled dsRNA was produced by Silencer siRNA Labeling kit-FAM according to the manufacturer's instructions.

## Pull-down assays

Pull-down assays were performed to detect Dcr-2–Loqs-PD interactions using His-tagged proteins purified from bacterial or insect cells. First, 1.25 µM His-tagged Loqs-PD and 0.6 µM untagged Dcr-2 were mixed and incubated on ice for 30 min. The protein mixture was then incubated with 15 µl Ni-NTA Agarose (Qiagen, 30210) in a total volume of 500 µl in the binding buffer (200 mM NaCl, 20 mM Tris pH 8.0, 5% glycerol, 20 mM imidazole) at 4 °C for 1 h with gentle rotation. After centrifugation at 500*i* for 1 min, the supernatant was removed and the beads were washed five times using wash buffer (200 mM NaCl, 20 mM Tris pH 8.0, 5% glycerol, 20 mM imidazole, 0.1% NP-40) by centrifugation, followed by SDS–PAGE analysis.

## In vitro dsRNA cleavage assays

Dicer-2–Loqs-PD cleavage assays were performed in cleavage buffer (50 mM HEPES pH 7.2, 100 mM NaCl, 1 mM DTT, 5 mM ATP) with dsRNA. Dcr-2–Loqs-PD and dsRNA were preincubated at 25 °C for 15 min, then added with ATP and MgCl₂ to a final concentration of 5 mM to start the reactions. The reactions were stopped with equal volume of 2× formamide loading buffer (95% formamide, 20 mM EDTA, 0.1% SDS, 0.005% xylene cyanol, 0.005% bromophenol blue). Samples were separated by 12% denaturing PAGE, visualized on Typhoon FLA-9000 (GE Healthcare) system.

## BS3/EDC-mediated cross-linking mass spectrometry

The purified complexes were incubated with 0.25 mM bis (sulfosuccinimidyl)suberate (BS3; Thermo Fisher Scientific, 21580) in the reaction buffer containing 50 mM HEPES pH 7.5, 80 mM NaCl and 5% glycerol at 25 °C for 2 h or 5 mM 1-ethyl-3-(3-dimethylaminopropyl) carbodiimide hydrochloride (EDC; Thermo Fisher Scientific, PG82073) in the reaction buffer containing 50 mM HEPES pH 7.2, 80 mM NaCl and 5% glycerol at 25 °C for 2 h. Cross-linked complexes were further purified to remove oligomer and glycerol by size-exclusion chromatography. The proteins (10 µg) were precipitated and digested for 16 h at 37 °C by trypsin at an enzyme-to-substrate ratio of 1:50 (w/w). The tryptic digested peptides were desalted and loaded on an in-house packed capillary reverse-phase C18 column (40 cm length, 100 µM ID × 360 µM OD, 1.9 µM particle size, 120 Å pore diameter) connected to an Easy LC 1200 system. The samples were analysed with a 120 min high-performance liquid chromatography gradient from 6% to 35% buffer B (buffer A: 0.1% formic acid in water; buffer B: 0.1% formic acid in 80% acetonitrile) at 300 nl min⁻¹. The eluted peptides were ionized and directly introduced into a Q-Exactive mass spectrometer using a nano-spray source. Survey full-scan MS spectra ($m/z = 300$–1,800) were acquired in the Orbitrap analyzer with resolution $r = 70,000$ at $m/z = 400$. Cross-linked peptides were identified and evaluated using pLink2 software[30].

## Cryo-EM sample preparation and data collection

We used the same specimen preparation and data collection method for all of the cryo-EM datasets. An aliquot of 4 µl of purified or reaction sample was applied to a custom-made graphene grid[31] (Quantifoil Au 1.2/1.3, 300 mesh), which were glow-discharged (in a Harrick Plasma system) for 10 s at middle level after 2 min evacuation. The grids were then blotted by a couple of 55 mm filter papers (Ted Pella) for 0.5 s at 22 °C and 100% humidity, then flash-frozen in liquid ethane using the FEI Vitrobot Mark IV. Cryo-EM data were collected on different Titan Krios electron microscopes, all of which were operated at 300 kV, equipped with a Gatan K3 direct electron detector and a Gatan Quantum energy filter. All data were automatically recorded using AutoEMation[32] or EPU (post-dicing state dataset) in counting mode and defocus values ranged from −1.5 µm to −2.0 µm. The other parameters of each dataset are provided in Extended Data Table 1.

## Image processing and 3D reconstruction

For all of the datasets, the image processing was adopted in similar steps. All of the raw dose-fractionated image stacks were 2× Fourier binned, aligned, dose-weighted and summed using MotionCorr2 (ref. [33]). The following steps were then processed in RELION (v.3.1)[34]. The contrast transfer function parameters were estimated using CTFFIND4 (ref. [35]). Approximately 2,000 particles were manually picked and 2D-classified to generate initial templates for automatic picking. A large number of particles were then automatically picked from raw micrographs on the basis of our templates. After one round of reference-free 2D

classification and several rounds of 3D classification, using the initial 3D reference models obtained by ab initio calculation in RELION v.3.1, particles from good 3D classes, with better overall structure features, were selected for 3D refinement. The final high-resolution homogeneous refinement was performed in CryoSPARC[36]. The resolutions were determined by gold-standard Fourier shell correlation. Local resolution distribution was evaluated using blocres command in the Bsoft software package[37]. The detailed image processing of each dataset is provided in Extended Data Figs. 2 and 3.

## Model building and refinement

The highest resolution EM density map of dimer status was used for initial model building, in which the quality of density was sufficient for de novo model building in COOT[38]. The initial model was separated into three parts (helicase-LoqsPD, DUF283 and other domains) and docked into EM 3D density maps of other states in Chimera[39] and then adjusted manually in ISOLDE[40] in Chimerax[41] and COOT. Finally, all of the models were refined against the EM map by PHENIX[42] in real space with secondary structure and geometry restraints. The final models were validated in PHENIX software package. The model statistics are summarized in Extended Data Table 1.

## Statistics and reproducibility

For Extended Data Fig. 1a–c, experiments were repeated at least three times. For Extended Data Figs. 1d, 2a,f, 3a,j, 9a and 10a–c, experiments were repeated at least twice.

## Reporting summary

Further information on research design is available in the Nature Research Reporting Summary linked to this paper.

## Data availability

The atomic coordinates and structure factors in apo and initial dsRNA-binding states, early- and mid-translocation states, active dicing and post-dicing states in this study have been deposited at the RCSB PDB and Electron Microscopy Data Bank (EMDB) under EMD accession codes EMD-32236, EMD-32237, EMD-32238, EMD-32239, EMD-32240 and EMD-32241, and PDB accession codes 7W0A, 7W0B, 7W0C, 7W0D, 7W0E and 7W0F, respectively. The PDB and EMDB codes are also listed in Extended Data Table 1. Uncropped gel images are provided in Supplementary Fig. 1. Other structures used in this study were retrieved from PDB with accession code 7VG2 (*At*DCL3), 2EZ6 (*Aa*RNase III), 2FFL (*Gi*Dicer), 5ZAK (*Hs*Dicer), 6LXD (*Hs*Drosha), 5F9H (*Hs*RIG-I) and 7ELE (*At*DCL1). The information of Dcr-2 and Loqs-PD is available in the UniProt database under accession codes A1ZAW0 and M9MRT5. Any other data or materials can be obtained from the corresponding authors on reasonable request.

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

**Acknowledgements** We thank Q. Liu for the Dcr-2 plasmid; J. Lei for assistance of data collection; and C. Peng, Y. Yin, C. Su and P. Wu for the mass spectrometry. We acknowledge the staff at the Tsinghua University Branch of the China National Center for Protein Sciences (Beijing) and Shuimu BioSciences for providing the cryo-EM facility support and the computational facility support. This research is supported by the National Natural Science Foundation of China (31971130 to J.M., 31825009 to H.-W.W., 32000849 to J.W. and 91940302 to Y.H.); the National Key R&D Program of China (2017YFA0503500 to X.L.); and the mRNA Innovation and Translation Center, Shanghai (to J.M.) and the Xplorer Prize (to H.-W.W.).

**Author contributions** J.M. and H.-W.W. conceived the study. J.H. and X.Y. initiated the project. S.S., J.W., J.M. and H.-W.W. designed experiments. S.S. and T.D. prepared the samples and performed the biochemical experiments, the mass spectrometry and analysed the data. S.S. and T.D. performed negative screening. S.S., J.W., N.L. and X.L. performed cryo-EM experiments and structure determination. J.M., S.S. and J.W. built the models. S.S. and J.W. refined the models. S.S., J.W., T.D., Y.H., J.M. and H.-W.W. analysed data. S.S., J.W., J.M. and H.-W.W. wrote the manuscript with input from the other authors.

**Competing interests** The authors declare no competing interests.

**Additional information**
**Correspondence and requests for materials** should be addressed to Hong-Wei Wang or Jinbiao Ma.

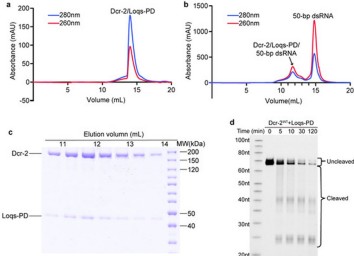

**Extended Data Fig. 1 | Protein purification and dsRNA-processing activity of Dcr-2–Loqs-PD. a–b**, Size exclusion chromatography results of Dcr-2–Loqs-PD and Dcr-2–Loqs-PD/50-bp dsRNA complex. **c**, SDS-PAGE of the protein fractions from **b**. **d**, Cleavage assays of Dcr-2–Loqs-PD$^{FL}$ (1.2 μM) with 50-bp dsRNA (1.2 μM), in the cleavage assay buffer (25 °C). Products were resolved on a 12% polyacrylamide denaturing gel. For gel source data, see Supplementary Figure 1.

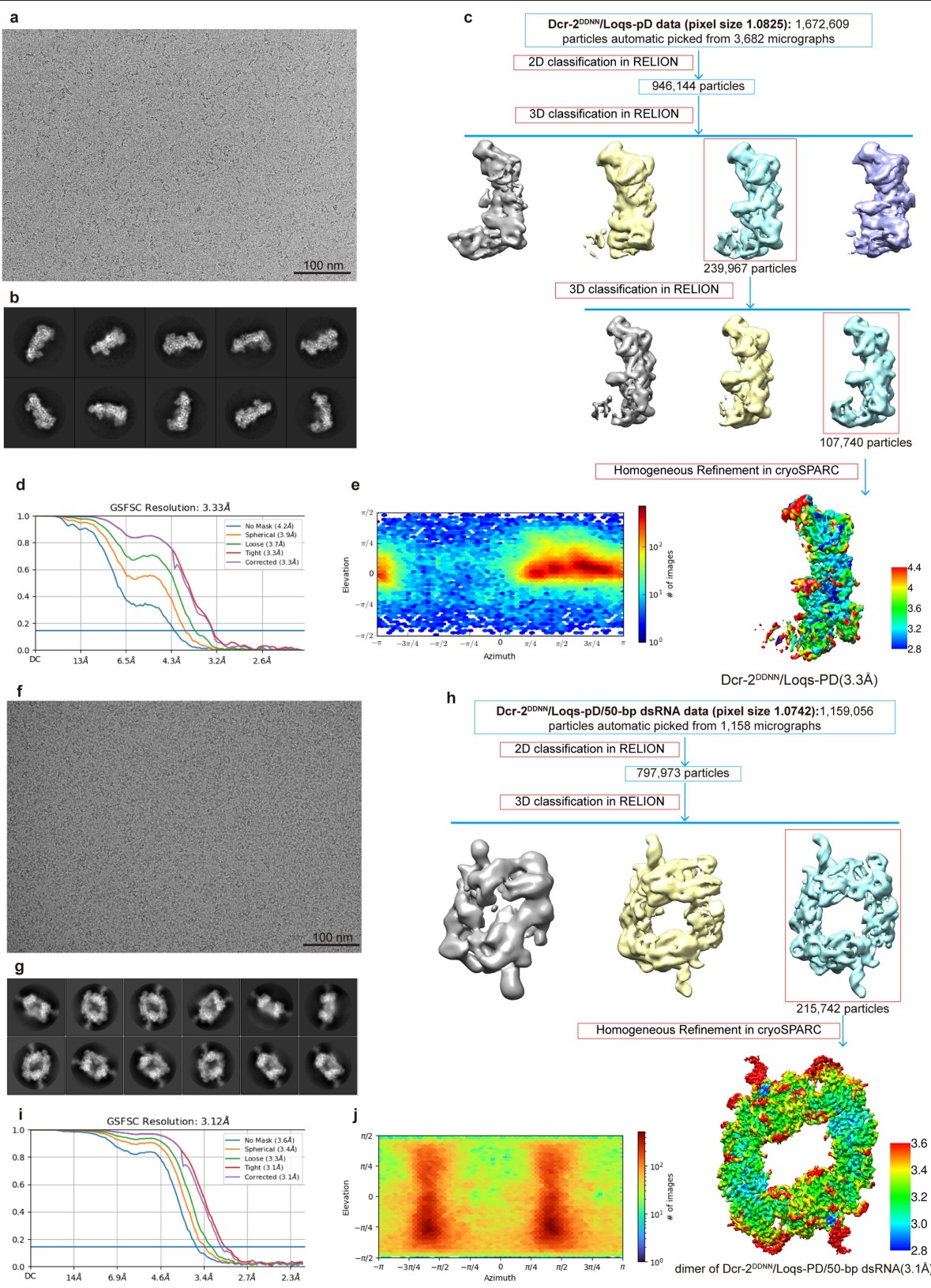

**Extended Data Fig. 2 | Cryo-EM image processing workflow of Dcr-2–Loqs-PD and its initial dsRNA-binding state. a**, A representative cryo-EM image of the Dcr-2–Loqs-PD complex in the apo-state. **b**, Representative views of 2D class averages of the apo-state complex. **c**, Flowchart of cryo-EM data processing of the Dcr-2–Loqs-PD complex. **d**, Gold-standard Fourier shell correlation (GSFSC) of the final map of Dcr-2–Loqs-PD. **e**, Particle distribution and final electron density map coloured according to the local resolution of Dcr-2–Lops-PD. **f–j**, Corresponding information as the apo-state for Dcr-2–Loqs-PD/dsRNA in the initial binding state.

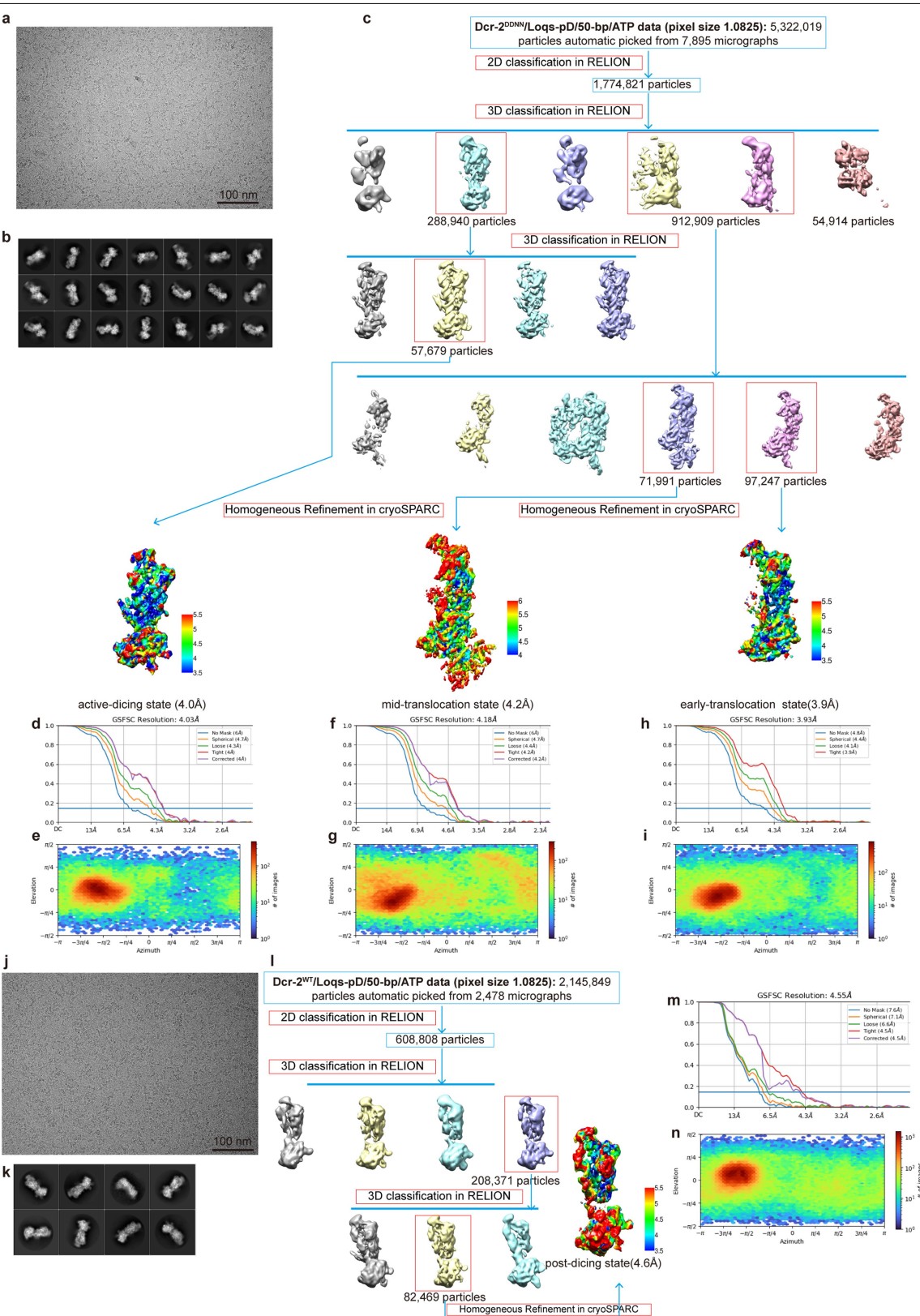

**Extended Data Fig. 3 | Cryo-EM image processing workflow of Dcr-2–Loqs-PD with dsRNA in the early- and mid-translocation, active dicing and post-dicing states. a–i**, Image processing information of Dcr-2^DDNN–

Loqs-PD/dsRNA in the translocation and active dicing states. Please note that we reconstructed three different maps from the same dataset. **j–n**, Image processing information of Dcr-2^WT–Loqs-PD/dsRNA in the post-dicing state.

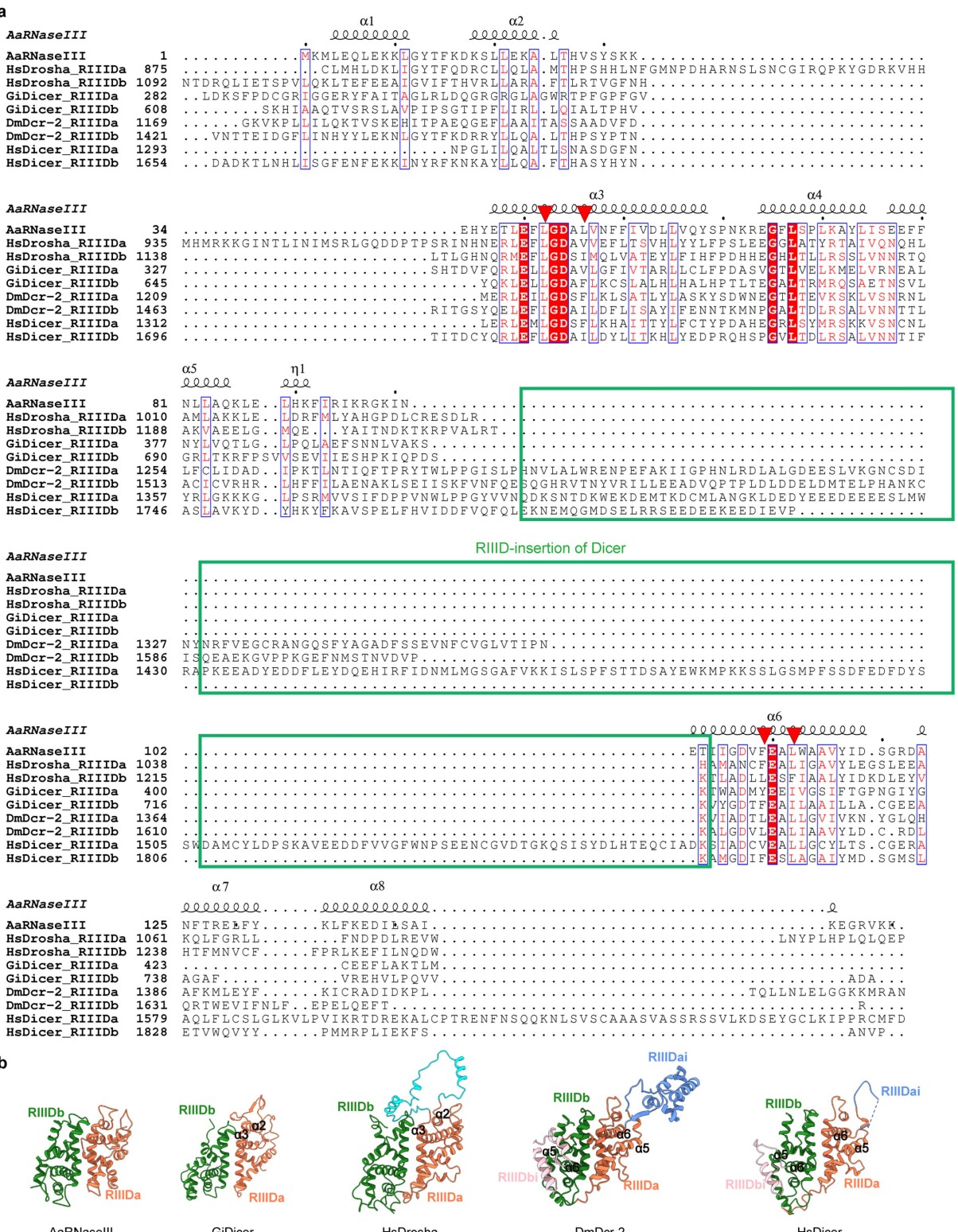

**Extended Data Fig. 4 | Sequence and Structure alignments of RIIIDs. a**, Sequence alignment of RIIIDs in different families of RNase III proteins (*Aa*RnaseIII, *Gi*Dicer, *Dm*Dcr-2, *Hs*Dicer, *Hs*Drosha). The secondary structure of *Aa*RNaseIII are at the top. The conserved residues interact with Mg2+ ions are labelled with red triangle. The RIIID insertion region is labelled by green box. *Aa, Aquifex aeolicus; Gi, Giardia intestinalis; Dm, Drosophila melanogaster; Hs, Homo sapiens*. **b**, Structures of RIIIDs corresponding to (**a**). *Aa*RNase III (PDB:

2EZ6), *Gi*Dicer (PDB: 2FFL), *Dm*Dcr-2 (apo state), *Hs*Dicer (PDB: 5ZAK), *Hs*Drosha (PDB: 6LXD) The helices adjacent to the insertion region are marked according to their appearance in *Gi*Dicer, *Dm*Dcr-2, *Hs*Dicer, and *Hs*Drosha. The insertion region of RIIIDs is coloured differently. RIIIDa insertion of Drosha is coloured in cyan. RIIIDa and RIIIDb insertions in Dicer family are coloured in cornflower-blue and pink, respectively.

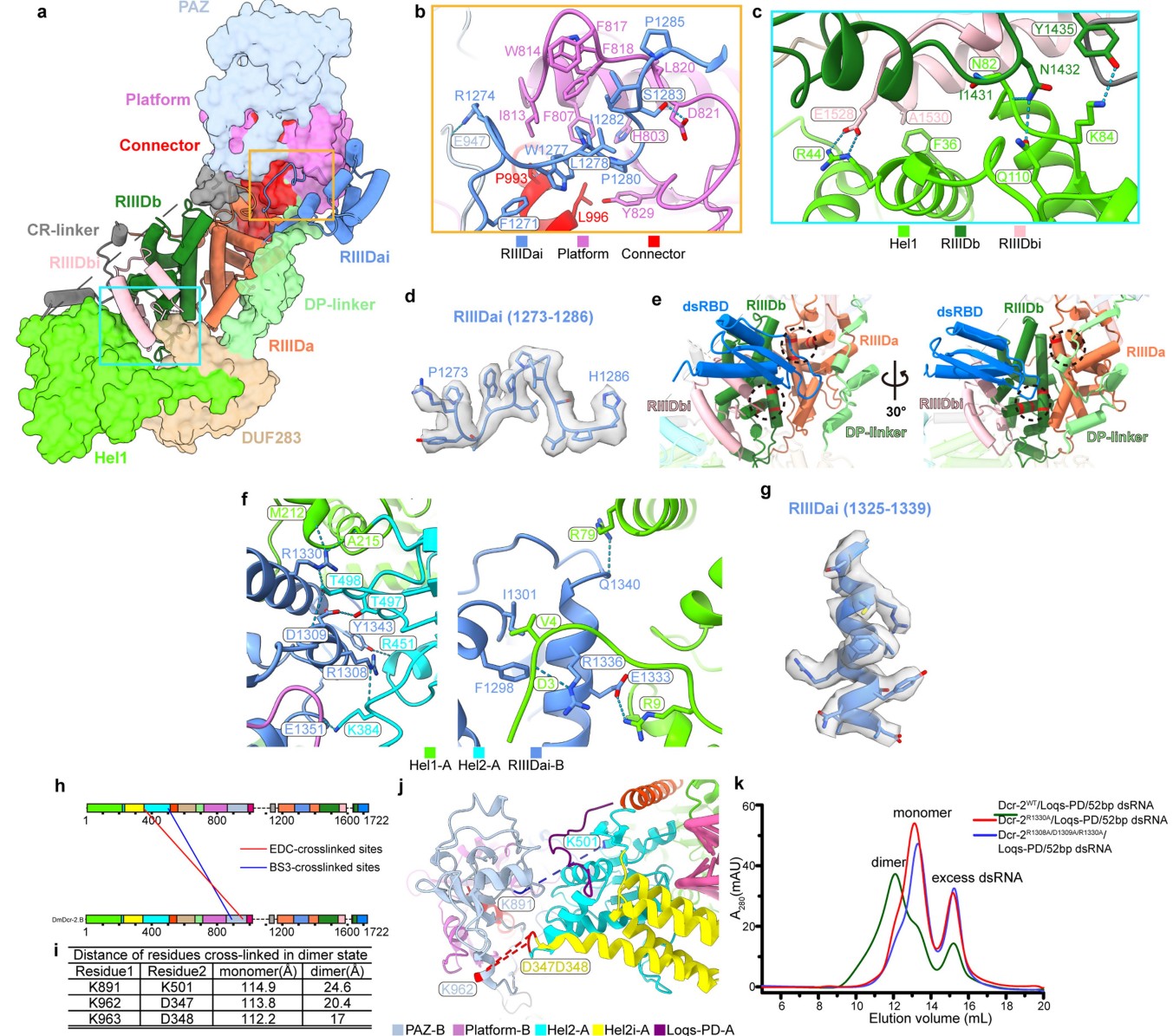

**Extended Data Fig. 5 | RIIID insertions interact with the regulatory regions of Dcr-2 and participate in dimer formation of the initial binding state.**
**a**, Overview of inter-region interactions of RIIIDai and RIIIDbi. Hel2, Hel2i, and C-terminal dsRBD are hidden in this panel. The core region except for the C-terminal dsRBD is shown in a cartoon model. Other parts of Dcr-2 is shown in transparent surface. **b**, Inter-region contacts of RIIIDai and the cap region, corresponding to the orange box in (**a**). The interactions between RIIIDai and the cap region are mainly hydrophobic. The residues involved in the interactions are shown as sticks. Cyan dashes represent the hydrogen bonds. **c**, Inter-region contacts of RIIIDb and RIIIDbi with the base region, corresponding to the cyan box in (**a**). The interactions between them are mainly hydrophilic. Residues and hydrogen bonds are marked as in (**c**). **d**, The EM density of aa 1273–1286 of RIIIDai domain with the fitted atomic model. **e**, Two different views of Dcr-2's core region in the apo-state. The RNase active centre

is marked with black dotted circles and the corresponding residues in the cartoon was labelled in red colour. **f**, Details of interactions at the dimer interface of the initial binding state. The interactions between RIIIDai and helicase domain are mainly hydrophilic. The residues involved in the interactions are shown as sticks. Cyan dashes represent the hydrogen bonds. **g**, The EM density of aa 1325–1339 of RIIIDai domain with fitted atomic models. **h**, Schematic summary of the inter-molecule cross-linking residues in the dimer of initial binding state from XL-MS result. The BS3 and EDC cross-linking sites are coloured in blue and red, respectively. **i**, The table of distance between cross-linking residues in monomer state or in dimer state. **j**, Close-up view of inter-molecule BS3 and EDC cross-linking residues in the initial binding state. The cross-linking site is coloured as in (**h**). **k**, Overlay view of the size exclusion chromatography results of Dcr-2[WT], Dcr-2[R1330A], and Dcr-2[R1308A/R1309A/R1330A] with Loqs-PD and dsRNA.

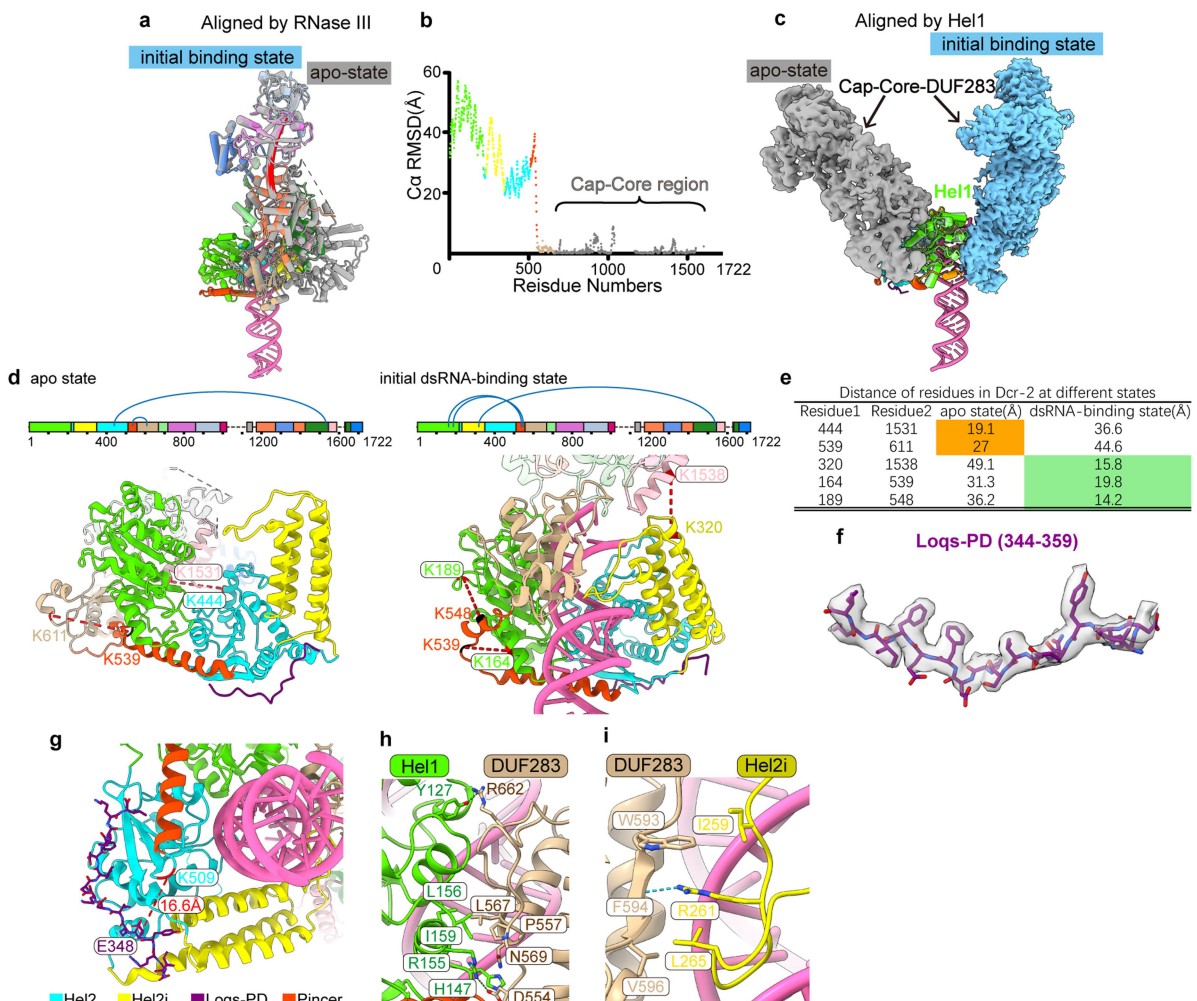

**a** Aligned by RNase III

initial binding state · apo-state

**b**

Cα RMSD(Å) vs Reisdue Numbers

Cap-Core region

**c** Aligned by Hel1

apo-state · Cap-Core-DUF283 · initial binding state

Hel1

**d** apo state

1 400 800 1200 1600 1722

K1531
K444
K611
K539

initial dsRNA-binding state

1 400 800 1200 1600 1722

K1538
K320
K189
K548
K539
K164

**e**

Distance of residues in Dcr-2 at different states

| Residue1 | Residue2 | apo state(Å) | dsRNA-binding state(Å) |
|---|---|---|---|
| 444 | 1531 | 19.1 | 36.6 |
| 539 | 611 | 27 | 44.6 |
| 320 | 1538 | 49.1 | 15.8 |
| 164 | 539 | 31.3 | 19.8 |
| 189 | 548 | 36.2 | 14.2 |

**f** Loqs-PD (344-359)

**g**

K509
16.6Å
E348

Hel2 · Hel2i · Loqs-PD · Pincer

**h** Hel1 · DUF283

Y127 · R662
L156 · L567
I159 · P557
R155 · N569
H147 · D554

**i** DUF283 · Hel2i

W593 · I259
F594 · R261
V596 · L265

**Extended Data Fig. 6 | The conformational change of the helicase domain induced by dsRNA binding. a**, Superposition of the apo state (grey) and the initial binding state (coloured), aligned by RIIIDs. **b**, The RMSD values of residues in (**a**) demonstrate the variation of the two state. The helicase domain is coloured as in Fig. 1a, and the cap-core region is coloured in grey. **c**, Superposition of the apo state (grey) and the initial binding state (coloured), aligned by the Hel1 domain. The DUF283-cap-core region is shown as EM density in the apo (grey) and initial binding (sky-blue) states, respectively. **d**, The BS3 cross-linking residues in the apo state (left panel) and the initial binding state (right panel) of the helicase domain. The cross-linked residues are

linked by blue line (upper panel) and red dashes (lower panel). **e**, Summary of distances between BS3 cross-linking residues in the apo state and the initial binding state. The cross-linking sites specific in the apo state are marked in the orange square. The cross-linking sites specific in the initial binding state are marked in the green square. **f**, The C-terminal region of Loqs-PD is shown in stick model fitted in the EM density in transparency. **g**, The EDC cross-linking residues between the helicase domain and C-terminal region of Loqs-PD. **h–i**, Inter-domain contacts of DUF283 with the Hel1 domain (**h**) and the Hel2i domain (**i**) in the initial binding state.

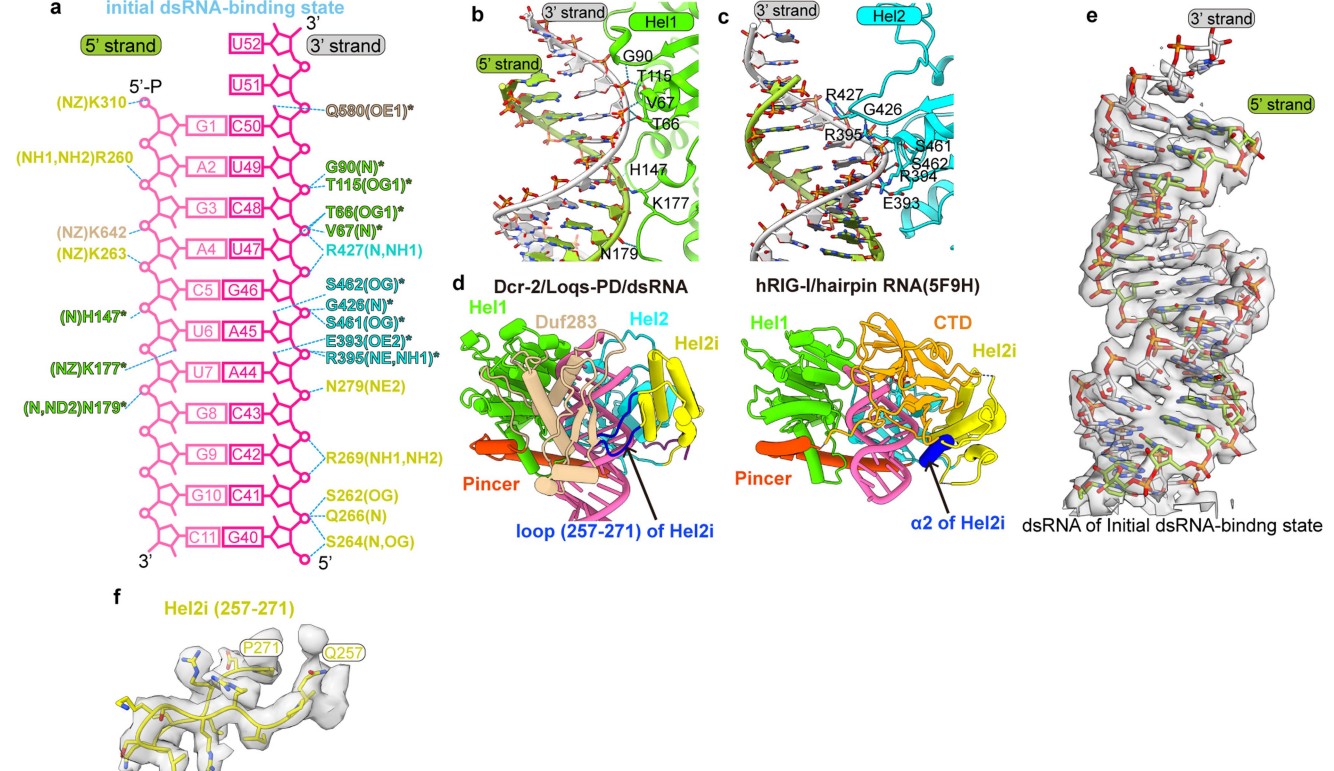

**Extended Data Fig. 7 | DsRNA recognition by the helicase domain in the initial binding state. a**, Schematic diagram showing the interactions between the helicase and dsRNA in the initial binding state. **b**–**c**, Close-up views of the Hel1-dsRNA (**b**), Hel2-dsRNA (**c**) interfaces. **d**, Overview of the helicase domain of Dcr-2 and *Hs*RIG-I (PDB: 5F9H) in the dsRNA-binding state. The special loop of Hel2i in Dcr-2 and the corresponding helix α2 in RIG-I are coloured in blue. **e**, The dsRNA of initial binding state is shown in stick model fitted in the EM density in transparency. **f**, The special loop of Hel2i is shown in stick model fitted in the EM density in transparency.

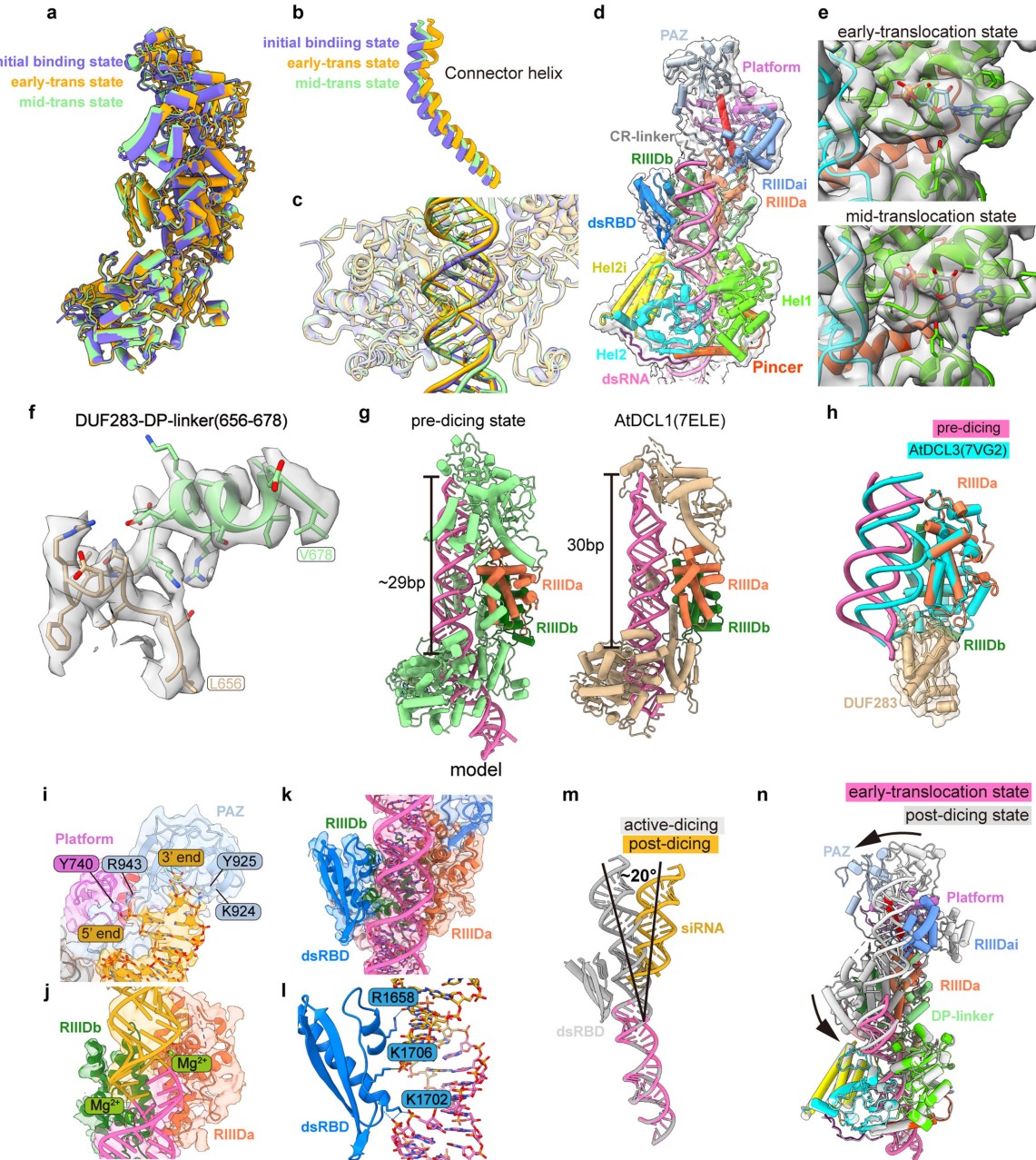

**Extended Data Fig. 8 | Structural detail and comparison of different states of Dcr-2–Loqs-PD/50-bp dsRNA complex. a**, Structural alignment of the initial binding, early-translocation and mid-translocation states of Dcr-2–Loqs-PD. The initial binding state is coloured in slate-blue. The early- and mid-translocation states are coloured in goldenrod and light-green, respectively. The colour scheme is used for (**a**–**c**). **b**, Structural comparison of the Connector helices, which represent the cap-core region positions in three states. The structures are aligned by their helicase domains. **c**, Superposition of the helicase domain in three states to display the trajectory of dsRNA. **d**, Overview of the mid-translocation state with transparent cryo-EM map. **e**, The density around the ATP-binding site in the early- and mid- translocation states. **f**, The interaction region of DUF283 and DP-linker (656–678) of initial binding state is shown in stick model fitted in the EM density in transparency. **g**, Structural comparison of the pre-dicing state and *At*DCL1/pre-miRNA complex (PDB: 7ELE), aligned by RIIIDs. **h**, Superposition the RIIIDs of

pre-dicing state and *At*DCL3/40-bp dsRNA complex. RIIIDs and dsRNA are shown in cartoon mode. DUF283 is shown in cartoon mode with transparent surface. **i**, Close-up view of the PAZ-Platform domain recognizing the terminus of dsRNA in the dicing state with transparent cryo-EM map. **j**, Close-up view of the processing centre of Dcr-2 in the active dicing state with transparent cryo-EM map. Two Mg$^{2+}$ ions are labelled. **k**, Overview of the interactions between the core region and dsRNA with transparent cryo-EM map. **l**, Close-up view of the C-terminal dsRBD/dsRNA interactions. **m**, Superimposition of RNA in the active dicing state (grey) and the post-dicing state (coloured). The C-terminal dsRBD of the active dicing state is shown in cartoon mode. The siRNA turned about 20 degrees after cleavage. **n**, Comparison of structures of the post-dicing state (light grey) and the early-translocation state (coloured) aligned by the helicase domain. Conformational changes from the post-dicing state to the early-translocation state are indicated by arrows.

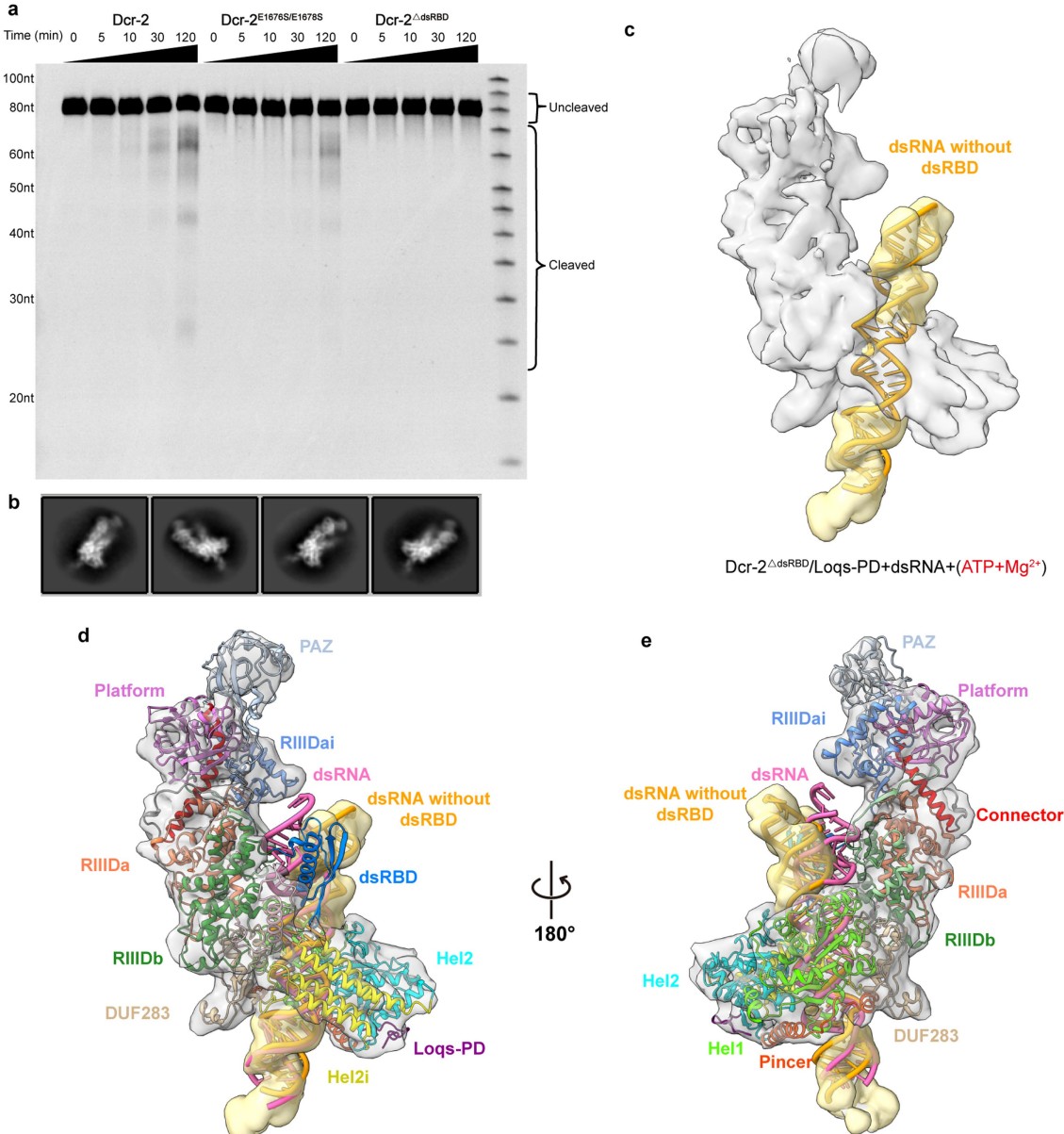

**Extended Data Fig. 9 | The C-terminal dsRBD of Dcr-2 is crucial for bending and efficient cleavage of dsRNA. a**, Cleavage assays were with 50-bp dsRNA (1.2 μM) with Dcr-2$^{WT}$, Dcr-2$^{E1676S/E1678S}$, and Dcr-2$^{\Delta dsRBD}$ (1.2 μM) with ATP, in the cleavage assay buffer (25 °C). Products were resolved on a 16% polyacrylamide denaturing gel. For gel source data, see Supplementary Figure 1. **b–c**, The 2D class averages and 3D reconstruction of Dcr-2$^{\Delta dsRBD}$/Loqs-PD+dsRNA in the presence of ATP and Mg$^{2+}$. The density and model of dsRNA is coloured in light yellow and orange, separately. **d–e**, Structural model of the mid-translocation state fitted in the 3D map of Dcr-2$^{\Delta dsRBD}$/Loqs-PD+dsRNA in two views.

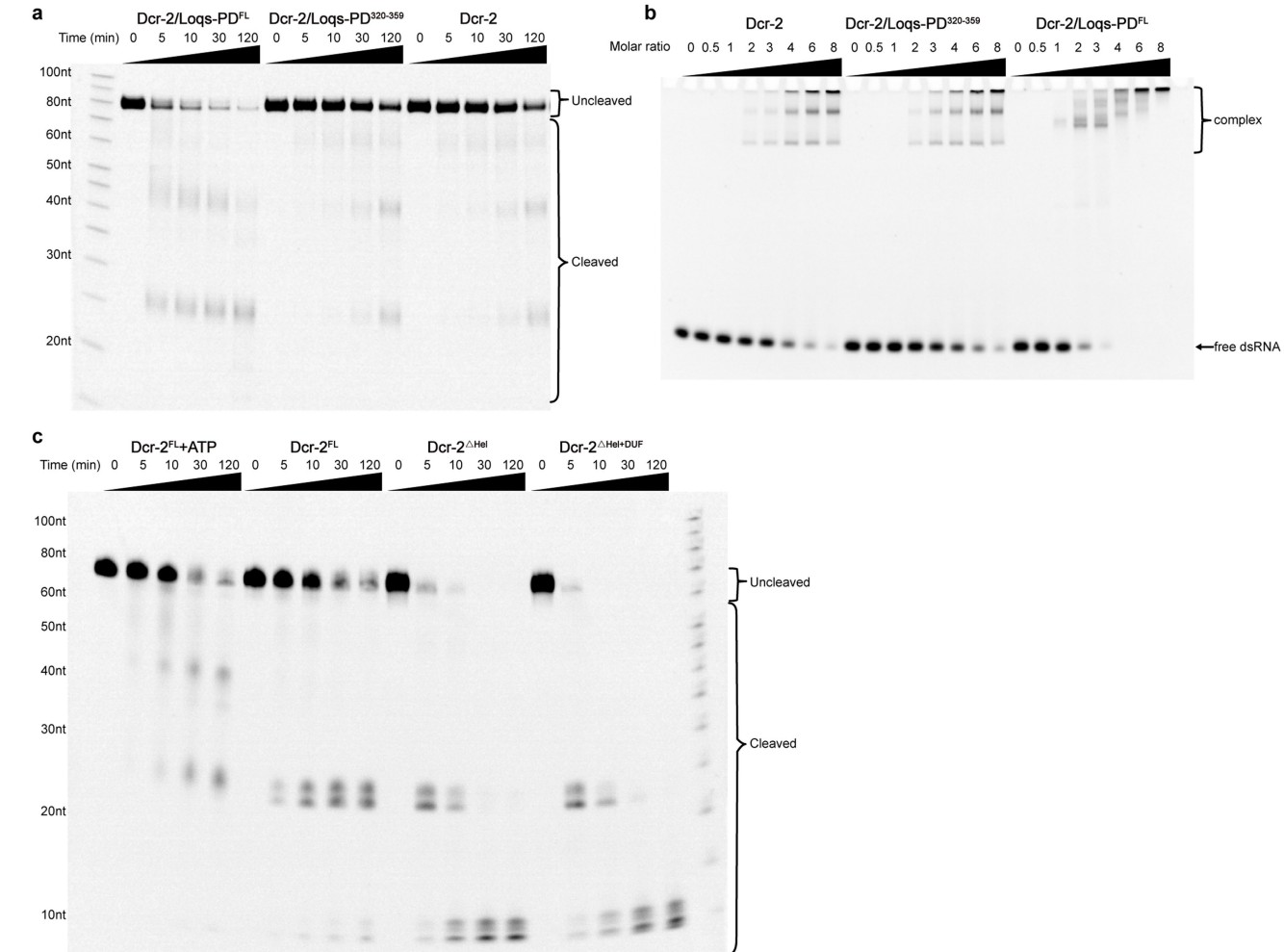

**Extended Data Fig. 10 | The base region plays a vital role in reducing nonspecific cleavage and dsRNA binding. a**, Cleavage assays of Dcr-2, Dcr-2–Loqs-PD[320–359], Dcr-2–Loqs-PD[FL] (1.2 μM) with 50-bp dsRNA (1.2 μM). **b**, EMSA experiment of Dcr-2, Dcr-2–Loqs-PD[320–359], Dcr-2–Loqs-PD[FL] with 50-bp dsRNA (0.2 μM). Molar ratio of protein and dsRNA are labelled. **c**, Cleavage assay of 50-bp dsRNA (1.2 μM) with Dcr-2[WT], Dcr-2[ΔHel], and Dcr-2[ΔHel-DUF] (3.6 μM). Substrate dsRNA and cleavage products are labelled by bracket arrows. For gel source data, see Supplementary Figure 1.

**Extended Data Table 1 | Cryo-EM data collection, processing, model refinement and validation statistics**

| | Apo state | Initial-binding | Early-translocation | Mid-translocation | Active-dicing | Post-dicing |
|---|---|---|---|---|---|---|
| PDB ID | 7W0B | 7W0A | 7W0C | 7W0D | 7W0E | 7W0F |
| EMDB ID | EMD-32237 | EMD-32236 | EMD-32238 | EMD-32239 | EMD-32240 | EMD-32241 |
| **Data collection and processing** | | | | | | |
| Microscope | | | Titan Krios | | | |
| Detector | | | Gatan K3 with GIF Quantum (20eV slit) | | | |
| CS (mm) | 2.7 | 2.7 | 2.7 | 2.7 | 2.7 | 0.01 |
| Magnification | 81K | 81K | 81K | 81K | 81K | 64K |
| Pixel size (Å) | 1.0825 | 1.0742 | 1.0825 | 1.0825 | 1.0825 | 1.08 |
| Electron dost (e$^-$/ Å$^2$) | 50(32 frames) | 50(32 frames) | 50(32 frames) | 50(32 frames) | 50(32 frames) | 50(32 frames) |
| Defocus range (μm) | -1.5 ~ -2.0 | -1.5 ~ -2.0 | -1.5 ~ -2.0 | -1.5 ~ -2.0 | -1.5 ~ -2.0 | -1.5 ~ -2.0 |
| Micrograph Number | 3,682 | 1,158 | 7,895 | 7,895 | 7,895 | 2,478 |
| **Reconstruction** | | | | | | |
| Software | | | RELION-3.1, cryoSPARC-3.2 | | | |
| Particles picked | 1,672,609 | 1,159,056 | 5,322,019 | 5,322,019 | 5,322,019 | 2,145,849 |
| Particles refinement | 57,076 | 215,742 | 97,247 | 71,991 | 57,679 | 82,469 |
| Symmetry | C1 | C2 | C1 | C1 | C1 | C1 |
| Resolution (Å) | 3.33 | 3.12 | 3.93 | 4.18 | 4.03 | 4.55 |
| Sharpening B-factor (Å$^2$) | 111.3 | 145.8 | 151.6 | 155.3 | 140.1 | 193.0 |
| **Refinement** | | | | | | |
| Software | | | PHENIX-1.19.2368 | | | |
| Model composition | | | | | | |
| Number of atoms | 12,997 | 26,902 | 13,663 | 20,322 | 14,849 | 14,097 |
| Protein residues | 1602 | 2,988 | 1,492 | 2,233 | 1557 | 1492 |
| Nucleotides | 0 | 126 | 72 | 104 | 104 | 94 |
| Ligand(Mg++/ADP) | 0/0 | 0/0 | 1/1 | 2/2 | 3/1 | 0/0 |
| B factors | 195.86/0/0 | 74.85/222.48/0 | 96.69/206.58/97.15 | 164.01/268.49/181.20 | 91.72/120.96/96.49 | 504.33/601.99/0 |
| Bonds RMSD | | | | | | |
| Bonds lengths (Å) | 0.003 | 0.004 | 0.002 | 0.002 | 0.002 | 0.002 |
| Bonds angles (°) | 0.576 | 0.525 | 0.547 | 0.584 | 0.553 | 0.579 |
| **Validation** | | | | | | |
| MolProbity score | 1.93 | 1.55 | 1.95 | 2.05 | 1.84 | 2.18 |
| Clash score | 11.64 | 6.92 | 14.09 | 15.63 | 10.67 | 21.42 |
| Rotamer outliers (%) | 0.00 | 0.00 | 0.00 | 0.05 | 0.07 | 0.89 |
| C-beta outliers (%) | 0.00 | 0.00 | 0.00 | 0.00 | 0.00 | 0.00 |
| Ramachandran plot | | | | | | |
| Favored (%) | 94.91 | 97.04 | 95.75 | 94.96 | 95.80 | 94.88 |
| Allowed (%) | 5.03 | 2.96 | 4.18 | 5.00 | 4.13 | 4.38 |
| Outlier (%) | 0.06 | 0.00 | 0.07 | 0.05 | 0.06 | 0.74 |
| Model vs. Data | | | | | | |
| CC mask/box | 0.79/0.84 | 0.80/0.82 | 0.73/0.86 | 0.80/0.91 | 0.74/0.85 | 0.40/0.73 |

# Reporting Summary

## Statistics

For all statistical analyses, confirm that the following items are present in the figure legend, table legend, main text, or Methods section.

| n/a | Confirmed | |
|---|---|---|
| ☐ | ☒ | The exact sample size ($n$) for each experimental group/condition, given as a discrete number and unit of measurement |
| ☒ | ☐ | A statement on whether measurements were taken from distinct samples or whether the same sample was measured repeatedly |
| ☐ | ☒ | The statistical test(s) used AND whether they are one- or two-sided<br>*Only common tests should be described solely by name; describe more complex techniques in the Methods section.* |
| ☒ | ☐ | A description of all covariates tested |
| ☒ | ☐ | A description of any assumptions or corrections, such as tests of normality and adjustment for multiple comparisons |
| ☐ | ☒ | A full description of the statistical parameters including central tendency (e.g. means) or other basic estimates (e.g. regression coefficient) AND variation (e.g. standard deviation) or associated estimates of uncertainty (e.g. confidence intervals) |
| ☐ | ☒ | For null hypothesis testing, the test statistic (e.g. $F$, $t$, $r$) with confidence intervals, effect sizes, degrees of freedom and $P$ value noted<br>*Give P values as exact values whenever suitable.* |
| ☒ | ☐ | For Bayesian analysis, information on the choice of priors and Markov chain Monte Carlo settings |
| ☒ | ☐ | For hierarchical and complex designs, identification of the appropriate level for tests and full reporting of outcomes |
| ☒ | ☐ | Estimates of effect sizes (e.g. Cohen's $d$, Pearson's $r$), indicating how they were calculated |

*Our web collection on statistics for biologists contains articles on many of the points above.*

## Software and code

Policy information about availability of computer code

| | |
|---|---|
| Data collection | We used AutoEMation (version 2.0) to collect cryo-EM datasets of Dcr-2/LoqsPD and dsRNA in apo, initial binding, translocation and dicing states, written by Dr. Jianlin Lei at Tsinghua University. Post-dicing state dataset was collected by EPU 2 of Thermo Fisher Scientific. |
| Data analysis | We used MotionCor2 (version 1.1.0) to correct the beam-induced motion of cryo-EM micrographs. The CTF values of these motion-corrected micrographs were determined by CTFFIND4 algorithm (version 4.15). We used Relion (version 3.1.3) and Cryo-SPARC (version 3.20) to perform image analysis and 3D reconstruction. Local resolution distribution was evaluated using  Bsoft(version 2.07) software package. We used COOT (version 0.9.4) for de novo model building and adjusted the model with COOT and ISOLDE (version 1.1).  All the models were refined against the EM map by PHENIX(version 1.19.2). The structural analysis was performed in UCSF Chimera (version 1.13.1) and ChimeraX (version 1.2.5). All these softwares are open-source except Cryo-SPARC, but it is free for educational users.<br>Cross-linked peptides were identified and evaluated using pLink2(version 2.3.9) software. |

For manuscripts utilizing custom algorithms or software that are central to the research but not yet described in published literature, software must be made available to editors and reviewers. We strongly encourage code deposition in a community repository (e.g. GitHub). See the Nature Portfolio guidelines for submitting code & software for further information.

## Data

Policy information about availability of data

All manuscripts must include a data availability statement. This statement should provide the following information, where applicable:

- Accession codes, unique identifiers, or web links for publicly available datasets
- A description of any restrictions on data availability
- For clinical datasets or third party data, please ensure that the statement adheres to our policy

The atomic coordinates and structure factors in apo and initial dsRNA binding states, early- and mid-translocation states, active-dicing and post-dicing states in this study have been deposited in the RCSB Protein Data Bank (PDB) and Electron Microscopy Data Bank (EMDB) under EMD accession codes 32236, 32237, 32238, 32239, 32240, 32241 and PDB ID codes 7W0A, 7W0B, 7W0C, 7W0D, 7W0E, 7W0F, respectively. The PDB and EMDB codes are also listed in Extended Data Table 1. For uncropped gel images, see Supplementary Figure 1. Other structures used in this study were retrieved from PDB with accession code 7VG2 (AtDCL3), 2EZ6 (AaRNase III), 2FFL (GiDicer), 5ZAK (HsDicer), 6LXD (HsDrosha), 5F9H (HsRIG-I), 7ELE (AtDCL1). The information of Dcr-2 and Loqs-PD could be found in Uniprot database with code A1ZAW0 and M9MRT5. Any other data or materials can be obtained from the corresponding author upon reasonable request.

# Field-specific reporting

Please select the one below that is the best fit for your research. If you are not sure, read the appropriate sections before making your selection.

☒ Life sciences ☐ Behavioural & social sciences ☐ Ecological, evolutionary & environmental sciences

For a reference copy of the document with all sections, see nature.com/documents/nr-reporting-summary-flat.pdf

# Life sciences study design

All studies must disclose on these points even when the disclosure is negative.

| | |
|---|---|
| Sample size | No sample size calculation was performed. For the cryo-EM maps, we ended up collecting data when we thought the structures were good enough or cannot be improved any more. For biochemical assays, we performed two to three replicates since the replications are successful. |
| Data exclusions | For cryo-EM reconstruction, particles grouped in bad classes with poorly defined features were excluded, because these particles were normally denatured or dissociated samples, which were harmful for high-resolution 3D reconstruction. |
| Replication | Our native gel shift assays, pull-down assays, and activity assays were performed in two to three independent replicates, all attempts at replication were successful. No data was excluded. |
| Randomization | Samples were allocated random, including the particle-motion and structural determination. |
| Blinding | For cryo-EM reconstruction, particles were randomly divided into two parts, and used for 3D structure determination. The consistence of structures generated by these two sub-datasets was used for the blinding test. For experiments other than cryo-EM, allowcation of samples into experiment groups was not performed in this study, therefore blinding was not relevant to our biochemical experiments. |

# Reporting for specific materials, systems and methods

We require information from authors about some types of materials, experimental systems and methods used in many studies. Here, indicate whether each material, system or method listed is relevant to your study. If you are not sure if a list item applies to your research, read the appropriate section before selecting a response.

## Materials & experimental systems

| n/a | Involved in the study |
|---|---|
| ☒ | ☐ Antibodies |
| ☐ | ☒ Eukaryotic cell lines |
| ☒ | ☐ Palaeontology and archaeology |
| ☒ | ☐ Animals and other organisms |
| ☒ | ☐ Human research participants |
| ☒ | ☐ Clinical data |
| ☒ | ☐ Dual use research of concern |

## Methods

| n/a | Involved in the study |
|---|---|
| ☒ | ☐ ChIP-seq |
| ☒ | ☐ Flow cytometry |
| ☒ | ☐ MRI-based neuroimaging |

## Eukaryotic cell lines

Policy information about cell lines

| | |
|---|---|
| Cell line source(s) | SF9, obtained from Invitrogen. |

| Authentication | The cell lines were obtained from commercial source and none of lines used were authenticated. |
| Mycoplasma contamination | Cell lines in this study were negative to mycoplasma by detection by PCR. |
| Commonly misidentified lines<br>(See ICLAC register) | No commonly misidentified cell lines were used in this study. |

