## [Peer Review File · Nature]

Manuscript Title: Structural insights into dsRNA processing by *Drosophila* Dicer-2/Loqs-PD

Reviewer Comments & Author Rebuttals

Reviewer Reports on the Initial Version:

Referees' comments:

Referee #1 (Remarks to the Author):

Su and Wang and co-workers present a structural analysis of dsRNA processing by the *Drosophila* Dcr-Loqs-PD complex. The study comprises 6 cryo-EM structures of Dcr2-Loqs-PD captured at various states of dsRNA translocation and cleavage. The result is a structural model for dsRNA translocation and cleavage by a Dicer enzyme that confirms prevailing hypotheses in the field, provides multiple important insights previously unknown, and is significantly more complete and detailed than any Dicer structural study published to date. Major findings include:

- Identification of the binding site for Loqs-PD on Dcr-2
- Initial dsRNA recognition is accompanied by a massive rearrangement of the Dcr-2 helicase/DUF283.
- Identification of the recognition site for substrate 5' phosphate groups in the Dcr-2 Hel2i domain
- The Dcr-2 Hel2i subdomain loop and dsRBD induce bends in bound dsRNAs to direct substrates towards the Dcr-2 PAZ domain and active sites.
- Structures of Dcr-2 in early and mid-translocation states, which lend irrefutable structural evidence to the translocation model for Dcr-2
- A model wherein DUF283 blocks dsRNA cleavage during translocation
- A structure of Dcr-2 in an active-dicing state, revealing an unexpected 10 Å separation of core and base modules
- A structure of Dcr-2 in a post-dicing state, revealing conformational changes associated with dsRNA cleavage.

Major comments, questions, and concerns:

The structural work is impressive and clearly an advance for the field. On the other hand, the proposed mechanistic models for how Dcr-2 achieves dicing fidelity remain untested. The study would thus benefit from a few focused biochemical experiments:

1) The authors propose the C-terminal dsRBD interacts with the DP linker and shields the RNase activity center in an auto inhibitory state to avoid cutting non-substrate RNAs. This is an interesting model, but stands in contrast to PMID: 18508075, which shows removal of the C-terminal dsRBD from human Dicer does not promote spurious dicing, suggesting RNase shielding is not necessary in human Dicer. Can the authors provide experimental evidence for this model in Dcr-2?

2) The authors also propose the interaction between DUF283 and RIIIDb prevent non-specific cleavage of dsRNA during translocation. This is another intriguing model for achieving fidelity in siRNA production. Does disrupting DUF283-RIIIDb interactions and/or extension of the DP-linker lead to spurious dsRNA cleavage? PMID: 18508075 also showed removal of helicase domains from human Dicer did not induce non-specific cleavage, but did increase dicing rates. Is it expected that Dcr-2 is different?

3) The most striking structural feature, in my opinion, is elongation of Dcr-2 in the active-dicing state followed by collapse to the post-dicing conformation associated with dsRNA cleavage. I am wondering if elongation of Dcr-2 in the active-dicing state is the result of tension in the complex that accumulates when the helicase translocates dsRNA that is bound to the PAZ domain. This model would provide a physical mechanism for Dcr-2 to connect dsRNA cleavage to proper recognition of the dsRNA end, enabling fidelity in siRNA production. Have the authors considered this model? If the observed elongated conformation is the active-dicing state, as proposed, does shortening the DP-linker to prevent elongation also inhibit dicing?

4) A related striking feature is that the dsRNA-binding cleft of the helicase base is not perfectly aligned with the dsRNA-binding cleft of the cap and core. Thus, as the authors show, either dsRNA or Dcr must bend for the two to engage. This is distinct from the popular simplistic model, which depicts dsRNA passing through Dicer unbent (for examples, see Fig. 7 of PMID: 22426548, Fig. 8 of PMID: 24488111, Fig. 7 of PMID: 25891075, Fig. 5 of PMID: 23661684, Fig. 6 of PMID: 21362554, Fig. 4 of PMID: 32903622). I am wondering if base-core misalignment inhibits dicing to help prevent non-specific dsRNA cleavage during translocation. Does a truncated Dcr-2, in which the helicase domains have been removed, cleave dsRNA more readily than full-length Dcr-2 (in the absence of ATP)?

5) The authors conclude that the overall conformation of Dcr-2 is relatively rigid during the translocation process. The data do not support this claim. Single particle analysis allows the authors to observe highly populated conformations, but many particles not conforming to these specific conformations are removed during processing. Thus, Dcr-2 may be dynamic during translocation (as might be expected for a translocating helicase), but may also often populate the observed "rigid" conformational state (possibly between translocation steps).

Minor suggestions and questions:

6) It is intriguing that the region of Loqs-PD visible in the reconstructions corresponds to the short C-terminal tail that distinguishes Loqs-PD from other forms of loquacious (see Fig. 1 of PMID: 19644447). Might it be worth noting that the interactions shown in Fig. 2c-g provide a structural basis for the specific requirement of Loqs-PD for endo-siRNA production in flies?

7) Line 142: "Loqs PD is a cofactor protein comprising of two dsRBDs responsible to recruit siRNA precursor substrates for Dcr 2." Shouldn't we expect to see some (weak) density corresponding to the Loqs-PD dsRBDs interacting with the ordered dsRNA in the initial-binding state reconstruction? I am curious because, considering the outstanding density for the dsRNA, a lack of density for Loqs-PD indicates Loqs-PD may function differently than currently believed. As the authors suggest, their reconstructions indicate the Loqs-PD C-terminal tail may prime the Dcr-2 helicase for dsRNA binding.

Is the C-terminal tail alone sufficient to promote dicing or are the dsRBDs also necessary?

8) It is surprising that the dsRBD is disordered in the post-dicing state, but this also makes sense mechanistically because the next step is to release the siRNA product. Is there a logical explanation for loss of dsRBD interactions after dicing? Does dicing disrupt the binding site for the dsRBD on the dsRNA?

9) Dimerization of the Dcr-2-Loqs-PD complex during initial dsRNA binding is intriguing and unexpected. What is the biological/mechanistic significance of the dimer? The Discussion section hints that maybe it is part of a response to low ATP levels. Do you know what ATP concentrations are necessary to disrupt the dimer?

10) The caption of Fig. 4a indicates the dsRNA is bent, but the dsRNA appears to be straight in the image. I also do not understand what the black axis represents. How can the authors define a helical axis (relative to Dcr-2) before Dcr-2 binding? I am similarly puzzled by the black axis in Fig. 4d, which does not align with any portion of the dsRNA. A clearer explication is needed here.

11) Fig. 3 shows substantial conformational change in helicase, including a large movement of DUF283 and the Pincer, upon dsRNA binding. This conformational change is intriguing, but not obvious in supplemental movie. Can the authors provide an animation specifically showing this conformational change? How does Loqs-PD relate to this conformational change?

Referee #2 (Remarks to the Author):

In this manuscript, Su and colleagues interrogate the mechanism of long dsRNA processing by fly *dcr-2* by resolving cryo-EM structures of *dcr-2* at different stages of dsRNA processing (apo, initial-binding, early-translocation, mid-translocation, active-dicing, and post-dicing states). This is by far the most comprehensive study of dsRNA processing by *dcr-2*, giving us new valuable insights on the molecular mechanisms by which other Dicer homologs and paralogs may recognize and cleave their substrates.

Comments:

1. Some statements the authors made on the function of interdomain interactions need to be validated experimentally or should otherwise remain hypothetical.

The authors stated that “the C-terminal dsRBD interacts with the DP-linker and shields the RNase activity center in an auto-inhibitory state, which may avoid cutting non-substrate RNAs”. I wonder if the authors can support this idea in vitro by examining whether *dcr-2* with one or more of these interactions disrupted (e.g., by introducing point mutations) process the dsRNA substrates at faster rates than the wild-type *dcr-2* does. A competitive assay can also be performed to see if the mutant *dcr-2* cleaves the dsRNA substrate at slower rates in the presence of non-substrate RNA.

2. The authors also stated in the abstract that “the interaction between DUF283 and RIIIDb domains blocks the access of dsRNA to the RNase active center and prevents the non-specific cleavage of dsRNA”. The authors can support this idea, again, by disrupting this interaction and see if it alters processing rates and generates shorter products.

3. Is the dsRBD critical for the dsRNA bending? Cryo-EM structures with dsRBD-deleted Dicer-2 may address this question.

Minor points:

1. Please compare the 5'/3' end recognition mechanisms (Figure 5b) with previous studies (e.g., Tian et al., *Molecular Cell*, 2014) and also see if they are conserved.

2. Typos

- On line 170, RIG-I-like receptors (RLR) → on line 154
- Extended Data Figure 9e is missing in the body
- On line 237, Extended Data Fig. 9e → 9f

Referee #3 (Remarks to the Author):

Su et al.

Structural insights into ATP-dependent processing of long double-stranded RNAs by *Drosophila* Dicer-2/Loqs-PD

In this study, the authors have used cryoEM to investigate the structural basis of Dicer-2/Loqs-PD function. They expressed full-length Dcr-2 and Loqs-PD and also a Dcr-2 catalytic mutant. In addition, a 50 bp dsRNA in the presence or absence of ATP was investigated. This panel of condition allowed for the selection of particles resembling different states of the Dcr-2 loading and cleavage cycle. The authors walk through these states and conformations and highlight the key structural features and changes. First, the details of ATP binding by the helicase domain as well as the interactions with a short C-terminal fragment of Loqs-PD are presented and the interaction with Loqs-PD was validated by mutating key residues suggested by the cryoEM structure. Second, a comparison of the structure in the apo state and an early, initial dsRNA binding state is presented. The resolution is high enough to clearly define RNA-protein contacts in addition to overall rearrangements of domains and local structures. The helicase domain cooperates with the DUF283 domain and assembles around the dsRNA. Third, the authors selected a number of particle classes, which they defined as mid-translocation and pre-dicing stages. Here, the overall structure appears to be rather rigid and not many rearrangements are observed. Interestingly, as long as the dsRNA is absent from the catalytic center, the DUF283 domain blocks the RIIIDb sub-domain and the authors speculate that this might prevent promiscuous cleavage of non-substrates. The definition of the pre-dicing state is extrapolated by superimposition with a published AtDCL3 structure in the dicing state. This led to the conclusion that the dsRNA is not yet close enough to the catalytic center in the pre-dicing stage.

Fourth, particles were also collected from dicing and post-dicing stages. ATP hydrolysis is required for the transition from pre- to active-dicing stages since four extra bps are pushed through the helix domain in order to reach the Platform-PAZ domains and the catalytic center. Finally, a post-dicing step is postulated where again larger rearrangements were observed.

This is a comprehensive study of the activity cycle of *Drosophila* Dcr-2 in complex with dsRNA and a short fragment of Loqs-PD. Although Dicer cryoEM structures have been reported on human and plant Dicers, this study goes beyond these structures and adds novel insights into our current understanding of Dicer function. I find the different particles in different stages particularly appealing. However, the model is mainly based on these structural snapshots and some of the conclusions might be too speculative without further analysis. Indeed, several steps have also been reported for the plant Dicer enzymes before. Nevertheless, this well written manuscript is an important contribution. Several issues that need to be clarified are listed below.

1. The authors have used full-length Loqs-PD and Dcr-2. However, only a very short C-terminal fragment of Loqs-PD is resolved in the complex. It is clear that the dsRBDs are flexible and thus there might not be clear densities in the particles. It is nevertheless somewhat unexpected that no contacts during the translocation state between Loqs-PD and the dsRNA are observed. This would suggest that the sole function of Loqs-PD would be the recruitment to Dcr-2 but is not needed for the cleavage cycle at all. Is the C-terminal fragment presented in Figure 2 important for the transformation of the apo complex to the initial binding state? Furthermore, it would be interesting to investigate if Loqs-PD is needed at all in such a reconstituted *in vitro* system.
2. The authors observed larger dimeric complexes of the initial-binding state. Without further biochemical and functional investigations (mutations of residues that would prevent dimerization), this could well be an *in vitro* artifact. This should be mentioned clearly also in the results section of the manuscript.
3. The title of the chapter describing the transformation to the active-dicing state claims “ATP-dependent conformational changes...” (end of page 11) is somewhat misleading. ATP hydrolysis is not investigated and this assumption is based on the fact that the helicase may have passed four bps more compared to the pre-dicing state. I agree that this could be a likely scenario but without further testing, such a claim appears premature.
4. Figure 5: lines 271-273: “the cleavage site in the post-dicing state is close to the dsRBD binding position in the active dicing state, resulting in the loss of dsRBD density in the map.” This statement is unclear. Why would this fact lead to loss of density?
5. Shouldn't the cryoEM grids contain a snapshot of all intermediates of the dicing cycle? For example, during translocation, there should be equally distributed particles covering translocation by single bps. However, very distinct stages were obtained or selected. Is there a reason for that? Are there structural features that would result in a longer dwelling time of Dicer in a particular position or conformation?

Response to reviewers:

(Reviewers' comments in italic and our responses in red)

Referee #1 (Remarks to the Author):

Su and Wang and co-workers present a structural analysis of dsRNA processing by the Drosophila Dcr-Loqs-PD complex. The study comprises 6 cryo-EM structures of Dcr2-Loqs-PD captured at various states of dsRNA translocation and cleavage. The result is a structural model for dsRNA translocation and cleavage by a Dicer enzyme that confirms prevailing hypotheses in the field, provides multiple important insights previously unknown, and is significantly more complete and detailed than any Dicer structural study published to date. Major findings include:

- *Identification of the binding site for Loqs-PD on Dcr-2*
- *Initial dsRNA recognition is accompanied by a massive rearrangement of the Dcr-2 helicase/DUF283.*
- *Identification of the recognition site for substrate 5' phosphate groups in the Dcr-2 Hel2i domain*
- *The Dcr-2 Hel2i subdomain loop and dsRBD induce bends in bound dsRNAs to direct substrates towards the Dcr-2 PAZ domain and active sites.*
- *Structures of Dcr-2 in early and mid-translocation states, which lend irrefutable structural evidence to the translocation model for Dcr-2*
- *A model wherein DUF283 blocks dsRNA cleavage during translocation*
- *A structure of Dcr-2 in an active-dicing state, revealing an unexpected 10 Å separation of core and base modules*
- *A structure of Dcr-2 in a post-dicing state, revealing conformational changes associated with dsRNA cleavage.*

We thank this reviewer's recognition of the significance and relevance of our work.

Major comments, questions, and concerns:

The structural work is impressive and clearly an advance for the field. On the other hand, the proposed mechanistic models for how Dcr-2 achieves dicing fidelity remain untested. The study would thus benefit from a few focused biochemical experiments:

1) The authors propose the C-terminal dsRBD interacts with the DP linker and shields the RNase activity center in an auto inhibitory state to avoid cutting non-substrate RNAs. This is

an interesting model, but stands in contrast to PMID: 18508075, which shows removal of the C-terminal dsRBD from human Dicer does not promote spurious dicing, suggesting RNase shielding is not necessary in human Dicer. Can the authors provide experimental evidence for this model in Dcr-2?

We thank the referee for the suggestions. We performed the following experiments and made interesting discoveries. We mutated residues of E1676 and E1678 of the C-terminal dsRBD at the interface of C-terminal dsRBD and DP-linker to serine residue (E1676S/E1678S) and compared the mutant's activity with that of Dcr-2^{WT}. The result showed that the cleavage ability of the mutant E1676S/E1678S in the presence of ATP was dramatically reduced (Fig. R1a). Moreover, removal of the C-terminal dsRBD (Dcr-2^{ΔdsRBD}) completely abolished Dcr-2's cleavage ability for dsRNA substrate in the presence of ATP. This indicates the critical role of the C-terminal dsRBD in the ATP-dependent processing of dsRNA substrate, probably by holding the dsRNA stem towards the PAZ domain during the translocation. This result is consistent with the conclusion that the C-terminal dsRBD is critical for the cleavage activity of Dcr-2. However, similar to human Dicer, removal of the C-terminal dsRBD of Dcr-2 does not promote spurious dicing, suggesting RNase shielding is also not necessary for Dcr-2. Therefore, we deleted the statement "auto inhibitory state to avoid cutting non-substrate RNAs" based on our current biochemical experiments in the new manuscript.

We have updated our manuscript with the new results in Extended Data Figure 10.

Fig. R1. a, Cleavage assays were with 50-bp dsRNA (1.2 μ M) with Dcr-2^{WT}, Dcr-2^{E1676S/E1678S}, and Dcr-2^{ΔdsRBD} (1.2 μ M) with ATP (25°C). Products were resolved on a 16% polyacrylamide

denaturing gel.

2) The authors also propose the interaction between DUF283 and RIIIDb prevent non-specific cleavage of dsRNA during translocation. This is another intriguing model for achieving fidelity in siRNA production. Does disrupting DUF283-RIIIDb interactions and/or extension of the DP-linker lead to spurious dsRNA cleavage?

As suggested, we made two mutants, one containing an insertion of four amino acids (GSGS) between K666 and E667 (DP-linker extension) and the other also containing a single mutation V622R at the interface of DUF283-RIIIDb interaction (disrupting interaction). In comparison with the cleavage activities of wildtype Dcr-2/Loqs-PD complex, both mutants exhibited significant enhancement of cleavage efficiency (Fig. R2). This suggests that weakening the interactions between DUF283 and RIIIDb and the extension of the DP-linker may allow easier conformational change from the pre-dicing state to the active-dicing state. We did not detect obvious spurious dsRNA cleavage for the mutants compared to the wild-type complex.

Fig. R2. Extension of the DP-linker causes higher efficiency of siRNA production. Cleavage assays were with 50-bp dsRNA (1.2 μ M) with Dcr-2^{WT}/Loqs-PD, Dcr-2^{666GSGS667}/Loqs-PD, and Dcr-2^{V622R-666GSGS667}/Loqs-PD (1.2 μ M) with ATP in the cleavage assay buffer (25°C). Products were resolved on a 12% polyacrylamide denaturing gel.

PMID: 18508075 also showed removal of helicase domains from human Dicer did not induce non-specific cleavage, but did increase dicing rates. Is it expected that Dcr-2 is different?

We prepared two Dcr-2 truncation mutants, Dcr-2^{ΔHel} (552-1722, removal of the helicase domain) and Dcr-2^{ΔHel-DUF} (665-1722, removal of the helicase and DUF283 domains), and compared the dsRNA cleavage activities of Dcr-2^{WT} with or without ATP, Dcr-2^{ΔHel}, and Dcr-2^{ΔHel-DUF} (Fig. R3). Similar to human Dicer, the truncation of Helicase domain did increase the dicing rates in the absence of ATP. However, compared to the more specific cleavage products from Dcr-2^{WT}, the truncated mutants produced shorter cleavage non-specific products around 11nt in the absence of ATP. These results suggest that Dcr-2 may use different means to recognize and cleave the dsRNA substrate in the absence of ATP, not involving the Helicase domain. Additional truncation of DUF283 did not further affect the cleavage activity. We have updated these results in the manuscript as new Extended Data Figure 11.

Fig. R3. Cleavage assay of 50-bp dsRNA (1.2 μM) with Dcr-2^{WT}, Dcr-2^{ΔHel}, and Dcr-2^{ΔHel-DUF} (3.6 μM). Substrate dsRNA and cleavage products are labeled by bracket arrows.

3) The most striking structural feature, in my opinion, is elongation of Dcr-2 in the active-dicing state followed by collapse to the post-dicing conformation associated with dsRNA cleavage. I am wondering if elongation of Dcr-2 in the active-dicing state is the result of tension in the complex that accumulates when the helicase translocates dsRNA that is bound

to the PAZ domain. This model would provide a physical mechanism for Dcr-2 to connect dsRNA cleavage to proper recognition of the dsRNA end, enabling fidelity in siRNA production. Have the authors considered this model? If the observed elongated conformation is the active-dicing state, as proposed, does shortening the DP-linker to prevent elongation also inhibit dicing?

We totally agree with this reviewer on the tension in the Dcr-2/Loqs-PD-dsRNA complex that accumulates when the helicase translocates on dsRNA. The tension built-up is probably due to the misalignment between the bottom module that binds to dsRNA through Helicase domain and the PAZ domain that binds to the terminal of dsRNA. We tried to test this model by generating three DP-linker shortened mutants (fragment of 668-671 replaced by GP, fragment of 669-677 replaced by SYVAIS, fragment of 669-677 replaced by GSGSGS) according to the reviewer's suggestion. Unfortunately, all three mutants could not be purified due to the extremely low expression level. However, as previously showed, the extension of DP-linker can lead to higher cleavage activity, probably due to the easier conformational change from the pre-dicing to the active dicing states (Fig. R2). We have updated our manuscript with more clear description of the physical mechanism as suggested by the reviewer.

4) A related striking feature is that the dsRNA-binding cleft of the helicase base is not perfectly aligned with the dsRNA-binding cleft of the cap and core. Thus, as the authors show, either dsRNA or Dcr must bend for the two to engage. This is distinct from the popular simplistic model, which depicts dsRNA passing through Dicer unbent (for examples, see Fig. 7 of PMID: 22426548, Fig. 8 of PMID: 24488111, Fig. 7 of PMID: 25891075, Fig. 5 of PMID: 23661684, Fig. 6 of PMID: 21362554, Fig. 4 of PMID: 32903622). I am wondering if base-core misalignment inhibits dicing to help prevent non-specific dsRNA cleavage during translocation. Does a truncated Dcr-2, in which the helicase domains have been removed, cleave dsRNA more readily than full-length Dcr-2 (in the absence of ATP)?

We appreciate the reviewer's discussion on the bending of dsRNA upon bound by Dcr-2, which is for the first time observed in multiple structures of Dcr-2/Loqs-PD in complex with dsRNA substrate. The structures of translocation states and activating-dicing state support the model that base-core misalignment prevent non-specific dsRNA cleavage during translocation. This model is also supported by the cleavage assay using the helicase-truncated Dcr-2 mutants, as shown in Fig. R3, which showed indeed a more readily cleavage of dsRNA than the full-length Dcr-2 in the absence of ATP.

5) The authors conclude that the overall conformation of Dcr-2 is relatively rigid during the

translocation process. The data do not support this claim. Single particle analysis allows the authors to observe highly populated conformations, but many particles not conforming to these specific conformations are removed during processing. Thus, Dcr-2 may be dynamic during translocation (as might be expected for a translocating helicase), but may also often populate the observed "rigid" conformational state (possibly between translocation steps).

Thank the reviewer to point this out. Although the overall conformations of Dcr-2 in initial-binding, early-translocation and mid-translocation states are quite similar, we could not exclude other conformations in highly dynamic transient states that could not be resolved by single particle analysis at high resolutions. In fact, some particles that only showed the dsRNA bound by Helicase-DUF283 domains were observed (Extended Data Fig. 3c, last class of 1st round 3D classification), which may represent the transient states. Similar structure was also observed in the paper of Science 2018. The structures obtained in this work represent the conformations at relatively low-energy, in which dsRNA substrates were bound by both bottom and core modules of Dcr-2. We have revised our manuscript to correct this statement.

Minor suggestions and questions:

6) *It is intriguing that the region of Loqs-PD visible in the reconstructions corresponds to the short C-terminal tail that distinguishes Loqs-PD from other forms of loquacious (see Fig. 1 of PMID: 19644447). Might it be worth noting that the interactions shown in Fig. 2c-g provide a structural basis for the specific requirement of Loqs-PD for endo-siRNA production in flies?*

We thank the reviewer for this informative suggestion. We have rewritten this part according to the reviewer's comment in the discussion section: "The interactions between C-terminal tail of Loqs-PD and Dcr-2 provide a structural basis for the specific requirement of Loqs-PD for endo-siRNA production (Ref: PMID: 19644447)."

7) *Line 142: "Loqs PD is a cofactor protein comprising of two dsRBDs responsible to recruit siRNA precursor substrates for Dcr 2." Shouldn't we expect to see some (weak) density corresponding to the Loqs-PD dsRBDs interacting with the ordered dsRNA in the initial-binding state reconstruction?*

From the two-dimensional averages of the initial binding state, we could not see density corresponding to the dsRBDs of Loqs-PD, indicating that: 1) Loqs-PD may bind RNA far away from the Dcr-2 molecule; 2) Loqs-PD may bind dsRNA without sequence specificity, resulting in the loss of density during after averaging.

I am curious because, considering the outstanding density for the dsRNA, a lack of density for Loqs-PD indicates Loqs-PD may function differently than currently believed. As the authors suggest, their reconstructions indicate the Loqs-PD C-terminal tail may prime the Dcr-2 helicase for dsRNA binding. Is the C-terminal tail alone sufficient to promote dicing or are the dsRBDs also necessary?

To answer this and other reviewer's question, we generated a construct only containing C-terminal tail of Loqs-PD (320-359), and measured the dsRNA-binding affinity by EMSA and dsRNA cleavage activity of Dcr-2 in the presence of different Loqs-PD constructs. The result showed that the C-terminal tail alone of Loqs-PD did not enhance Dcr-2 cleavage activity nor binding affinity (Fig. R4), suggesting that the dsRBD domains of Loqs-PD are necessary for its cofactor activity. We have updated our manuscript with these results in the Extended Data Figure 11.

Fig. R4. C-terminal tail of Loqs-PD alone has no effect on dsRNA-binding and siRNA production. **a**, Cleavage assay of Dcr-2, Dcr-2/Loqs-PD³²⁰⁻³⁵⁹, Dcr-2/Loqs-PD^{FL} (1.2 μM) with 50-bp dsRNA (1.2 μM). **b**, EMSA experiment of Dcr-2, Dcr-2/Loqs-PD³²⁰⁻³⁵⁹, Dcr-2/Loqs-PD^{FL} with 50-bp dsRNA (0.2 μM). Molar ratios of protein and dsRNA are labeled.

8) It is surprising that the dsRBD is disordered in the post-dicing state, but this also makes sense mechanistically because the next step is to release the siRNA product. Is there a logical explanation for loss of dsRBD interactions after dicing? Does dicing disrupt the binding site for the dsRBD on the dsRNA?

We thank the reviewer for this suggestion. We added the following explanation in the revised manuscript (page13, Lines 281-285) : "The cleavage of dsRNA disrupts the binding

site for the C-terminal dsRBD in active-dicing state (Fig. 5d-f, l), and probably results in the dsRBD becoming a more flexible and losing its density in the averaged EM map of post-dicing state (Fig. 5k, m), which may be also favorable to the release of cleaved siRNA products (Fig. 5l and Extended Data Video 1).”

9) Dimerization of the Dcr-2-Loqs-PD complex during initial dsRNA binding is intriguing and unexpected. What is the biological/mechanistic significance of the dimer? The Discussion section hints that maybe it is part of a response to low ATP levels. Do you know what ATP concentrations are necessary to disrupt the dimer?

The dimer of initial-binding state complex was obtained in a condition without ATP. We calculated the statistics of dimer ratio at different ATP concentrations by negative staining EM and noticed a negative correlation of the dimer ratio with ATP concentration (Fig. R5). The dimer completely dissociated at 2 mM ATP. Because a normal cellular ATP concentration is maintained in the range of 1 to 10 mM, we cannot conclude at current stage whether the dimer is an in vitro artifact or physiological relevant. We will keep studying this interesting phenomenon in our future work.

Fig. R5. Negative staining EM results of Dcr-2/Loqs-PD/dsRNA complex without (a) or with 0.5 mM ATP(b). The upper panels are representative original micrographs. The lower panels are 2D classification results. Classes of dimer particles are marked by red boxes. (c) The statistics of dimer ratio of Dcr-2/Loqs-PD/dsRNA complex with different concentration of ATP.

10) The caption of Fig. 4a indicates the dsRNA is bent, but the dsRNA appears to be straight in the image. I also do not understand what the black axis represents. How can the authors define a helical axis (relative to Dcr-2) before Dcr-2 binding? I am similarly puzzled by the black axis in Fig. 4d, which does not align with any portion of the dsRNA. A clearer explication is needed here.

We thank the reviewer for this suggestion. We have redrawn Fig. 4 and rewritten the related legend of the Fig. 4a in the revised manuscript: “The axis of idea A-form dsRNA extended from outside of Helicase domain is shown in black line. The helical axis direction of Helicase domain bound dsRNA is shown in red arrow.”, and added the legend of the Fig. 4c: “The axis of idea A-form dsRNA extended from outside of Helicase domain is shown in black line. The helical axis of C-terminal dsRBD domain-bound dsRNA is shown in red arrow.” The idea A-form dsRNA are shown in the black and white in Fig. R6b and R6d.

Fig. R6. The bent angle of dsRNA is measured by comparing with ideal A-form dsRNA. **a-b**, Distortion of dsRNA in the early-translocation state. The axis direction of Dcr-2-binding dsRNA and ideal A-form dsRNA is shown in red arrow and black line, respectively. **c-d**, Distortion of dsRNA in mid-translocation state is shown in the same mode as in **a-b**.

11) Fig. 3 shows substantial conformational change in helicase, including a large movement of DUF283 and the Pincer, upon dsRNA binding. This conformational change is intriguing, but not obvious in supplemental movie. Can the authors provide an animation specifically showing this conformational change? How does Loqs-PD relate to this conformational change?

We made a new video for this conformational change of the helicase domain from the apo state to the initial-binding state (Extended Data Video 2). As shown in **Fig. R4** and reported earlier, the only function of Loqs-PD is to help Dcr-2 binding dsRNA substrates, and Loqs-PD is not related to the conformational change, since we cannot observe densities of Loqs-PD's dsRBDs in our apo and initial-binding state.

Referee #2 (Remarks to the Author):

In this manuscript, Su and colleagues interrogate the mechanism of long dsRNA processing by fly *dcr-2* by resolving cryo-EM structures of *dcr-2* at different stages of dsRNA processing (apo, initial-binding, early-translocation, mid-translocation, active-dicing, and post-dicing states). This is by far the most comprehensive study of dsRNA processing by *dcr-2*, giving us new valuable insights on the molecular mechanisms by which other Dicer homologs and paralogs may recognize and cleave their substrates.

We thank the reviewer's recognition of the significance of our work.

Comments:

1. *Some statements the authors made on the function of interdomain interactions need to be validated experimentally or should otherwise remain hypothetical.*

The authors stated that "the C-terminal dsRBD interacts with the DP-linker and shields the RNase activity center in an auto-inhibitory state, which may avoid cutting non-substrate RNAs". I wonder if the authors can support this idea in vitro by examining whether dcr-2 with one or more of these interactions disrupted (e.g., by introducing point mutations) process the dsRNA substrates at faster rates than the wild-type dcr-2 does. A competitive assay can also be performed to see if the mutant dcr-2 cleaves the dsRNA substrate at slower rates in the presence of non-substrate RNA.

Please see our response to Referee #1.

2. *The authors also stated in the abstract that "the interaction between DUF283 and RIIIDb domains blocks the access of dsRNA to the RNase active center and prevents the non-specific cleavage of dsRNA". The authors can support this idea, again, by disrupting this interaction and see if it alters processing rates and generates shorter products.*

Please see our response to Referee #1.

3. *Is the dsRBD critical for the dsRNA bending? Cryo-EM structures with dsRBD-deleted Dicer-2 may address this question.*

We thank the reviewer for this suggestion. In order to answer this question, we purified the C-terminal dsRBD-deleted Dcr-2/Loqs-PD complex, mixed it with dsRNA substrate and performed cryo-EM analysis of the complex. From the 2D classification averages and a 3D reconstruction (Fig. R7a), we obtained a structure in which the dsRNA duplex threads through the helicase domain to reach a similar position as in the mid-translocation state (Fig. R7b). However, compared to the structure of wild-type Dcr-2/Loqs-PD/dsRNA complex, the dsRNA does not have obvious bending in the C-terminal dsRBD-deleted Dcr-2/Loqs-PD/dsRNA structure (Fig. R7c-d). This result confirmed that the dsRBD is critical for the dsRNA bending. We have updated this result into the main text and the Extended Data Figure 10.

Fig. R7. The C-terminal dsRBD of Dcr-2 is crucial for the dsRNA bending. **a-b**, 2D class averages and 3D reconstruction of Dcr-2^{ΔdsRBD}/Loqs-PD+dsRNA in the presence of ATP and Mg²⁺. The density of dsRNA is colored in light yellow. **c-d**, Structure of the mid-translocation state fitted in the 3D map of Dcr-2^{ΔdsRBD}/Loqs-PD+dsRNA in two views. There is no bending in the light-yellow dsRNA density.

Minor points:

1. Please compare the 5'/3' end recognition mechanisms (Figure 5b) with previous studies (e.g., Tian et al., *Molecular Cell*, 2014) and also see if they are conserved.

We thank the reviewer for the suggestion. However, due to the relative low resolution of the structures in the active-dicing (4.0 Å) and post-dicing states (4.6 Å), the side chains involved in the 5'/3' end recognition are not well defined. Therefore, we cannot make confident comparison with the structure of HsDicer (PDB: 4NH5, 4NGD). However, based on the current structure model of active-dicing state, we made a rough comparison and noticed that the end recognition mechanisms are not very conserved. The 3' end recognition is partially conserved, while the 5' end recognition mechanism is not conserved (Fig. R8).

Fig. R8. a, The 5' end recognition of DmDcr-2 (left panel) and HsDicer (right panel). **b**, The 3' end recognition of DmDcr-2 (left panel) and HsDicer (right panel).

2. Typos

- On line 170, RIG-I-like receptors (RLR) → on line 154
- Extended Data Figure 9e is missing in the body
- On line 237, Extended Data Fig. 9e → 9f

We have carefully proofread and corrected typos in the revised manuscript.

Referee #3 (Remarks to the Author):

Su et al.

Structural insights into ATP-dependent processing of long double-stranded RNAs by Drosophila Dicer-2/Loqs-PD

In this study, the authors have used cryoEM to investigate the structural basis of Dicer-2/Loqs-PD function. They expressed full-length Dcr-2 and Loqs-PD and also a Dcr-2 catalytic mutant. In addition, a 50 bp dsRNA in the presence or absence of ATP was investigated. This panel of condition allowed for the selection of particles resembling different states of the Dcr-2 loading and cleavage cycle. The authors walk through these states and conformations and highlight the key structural features and changes. First, the details of ATP binding by the helicase domain as well as the interactions with a short C-terminal fragment of Loqs-PD are presented and the interaction with Loqs-PD was validated by mutating key residues suggested by the cryoEM structure. Second, a comparison of the structure in the apo state and an early, initial dsRNA binding state is presented. The resolution is high enough to clearly define RNA-protein contacts in addition to overall rearrangements of domains and local structures. The helicase domain cooperates with the DUF283 domain and assembles around the dsRNA. Third, the authors selected a number of particle classes, which they defined as mid-translocation and pre-dicing stages. Here, the overall structure appears to be rather rigid and not many rearrangements are observed. Interestingly, as long as the dsRNA is

absent from the catalytic center, the DUF283 domain blocks the RIIIDb sub-domain and the authors speculate that this might prevent promiscuous cleavage of non-substrates. The definition of the pre-dicing state is extrapolated by superimposition with a published AtDCL3 structure in the dicing state. This led to the conclusion that the dsRNA is not yet close enough to the catalytic center in the pre-dicing stage. Fourth, particles were also collected from dicing and post-dicing stages. ATP hydrolysis is required for the transition from pre- to active-dicing stages since four extra bps are pushed through the helix domain in order to reach the Platform-PAZ domains and the catalytic center. Finally, a post-dicing step is postulated where again larger rearrangements were observed.

This is a comprehensive study of the activity cycle of *Drosophila* Dcr-2 in complex with dsRNA and a short fragment of Loqs-PD. Although Dicer cryoEM structures have been reported on human and plant Dicers, this study goes beyond these structures and adds novel insights into our current understanding of Dicer function. I find the different particles in different stages particularly appealing. However, the model is mainly based on these structural snapshots and some of the conclusions might be too speculative without further analysis. Indeed, several steps have also been reported for the plant Dicer enzymes before. Nevertheless, this well written manuscript is an important contribution. Several issues that need to be clarified are listed below.

We thank the reviewer to recognize the significance of our work.

1. The authors have used full-length Loqs-PD and Dcr-2. However, only a very short C-terminal fragment of Loqs-PD is resolved in the complex. It is clear that the dsRBDs are flexible and thus there might not be clear densities in the particles. It is nevertheless somewhat unexpected that no contacts during the translocation state between Loqs-PD and the dsRNA are observed. This would suggest that the sole function of Loqs-PD would be the recruitment to Dcr-2 but is not needed for the cleavage cycle at all. Is the C-terminal fragment presented in Figure 2 important for the transformation of the apo complex to the initial binding state? Furthermore, it would be interesting to investigate if Loqs-PD is needed at all in such a reconstituted *in vitro* system.

Please see our response to Referee #1.

2. The authors observed larger dimeric complexes of the initial-binding state. Without further biochemical and functional investigations (mutations of residues that would prevent dimerization), this could well be an *in vitro* artifact. This should be mentioned clearly also in the results section of the manuscript.

We thank the reviewer for this suggestion. We have mutated the residues involved in dimerization and found the dimer was dissociated (**Extended Fig 7c**). We have also changed the related content in the revised manuscript (page6, Lines 119-122) : “The dimerization interface involves RIIDai domain of one Dcr-2 and Hel2 domain of the other Dcr-2, and mutation of residues in the dimerization interface results in dissociation of dimer (Fig. 1c and Extended Data Fig. 7).” Please also see our response to Referee #1.

3. The title of the chapter describing the transformation to the active-dicing state claims “ATP-dependent conformational changes...” (end of page 11) is somewhat misleading. ATP hydrolysis is not investigated and this assumption is based on the fact that the helicase may have passed four bps more compared to the pre-dicing state. I agree that this could be a likely scenario but without further testing, such a claim appears premature.

We thank the reviewer for this suggestion. We have changed the title of the chapter to: “Conformational change of Dcr-2 in the transition to active-dicing state” according to this reviewer’s comment.

4. Figure 5: lines 271-273: “the cleavage site in the post-dicing state is close to the dsRBD binding position in the active dicing state, resulting in the loss of dsRBD density in the map.” This statement is unclear. Why would this fact lead to loss of density?

We thank the reviewer for this suggestion. We added the following explanation in the revised manuscript (page13, Lines 281-285) : “The cleavage of dsRNA disrupts the binding site for the C-terminal dsRBD in active-dicing state (Fig. 5d-f, l), and probably results in the dsRBD becoming a more flexible and losing its density in the averaged EM map of post-dicing state (Fig. 5k, m), which may be also favorable to the release of cleaved siRNA products (Fig. 5l and Extended Data Video 1).”

5. Shouldn’t the cryoEM grids contain a snapshot of all intermediates of the dicing cycle? For example, during translocation, there should be equally distributed particles covering translocation by single bps. However, very distinct stages were obtained or selected. Is there a reason for that? Are there structural features that would result in a longer dwelling time of Dicer in a particular position or conformation?

We agree that the cryo-EM grids should contain a snapshot of all intermediates of the dicing cycle. But only high-populated states with defined and stable conformations can be classified into meaningful 2D and 3D averages using single particle cryo-EM analysis.

TSINGHUA UNIVERSITY

School of Life Sciences

Therefore, the structural states that we solved represent distinct conformations with some particular interactions, such as dsRBD binding dsRNA in the mid-translocation state. In addition, in order to improve the resolution of the structures, we focused more on the rigid parts of the molecules during the data processing. Thus, only a subset of relatively stable states, such as the early- and mid-translocation states, were obtained from data processing and 2D/3D average. There must be some more transient states with much less particle images representing other steps not solved by the current method. We will investigate the reason why the captured states are more stable than others and try to reveal other intermediate states in our future work.

Reviewer Reports on the First Revision:

Referees' comments:

Referee #1 (Remarks to the Author):

The authors addressed my previous concerns with thoughtfulness and diligence. I find the revised manuscript to be stronger than the original submission. Specifically, the authors disproved their original hypothesis of a role for the C-terminal dsRBD shielding the RNase active site and instead discovered the dsRBD is necessary for dicing activity. Taking this observation a step further, the authors provide new structural data showing the dsRBD is necessary for dsRNA bending, providing a functional connection between dsRNA bending and dicing activity. Additionally, the authors show that extending the DP-linker leads to greater dicing activity. Combined, these results further strengthen the exciting, and now quite compelling, model of a role for tension within the Dicer-dsRNA complex driving conformational change and dicing activity throughout the catalytic cycle. I have nothing further to add except to congratulate and thank the authors for this substantial addition to the field.

Referee #2 (Remarks to the Author):

The authors have adequately addressed my comments and the manuscript has been improved. I support the publication of this manuscript.

Referee #3 (Remarks to the Author):

In the revised version of their manuscript, Su et al. have addressed all points that I had raised on the previous version. Unclear points were clarified and better described. The authors have adequately responded to my comments and therefore I am satisfied.